# Research on an improved RT-DETR-based model for rice disease detection

Yaojun Zhang[1], Changqiang Shen [iD][1]*, Ying Xiong[2]

1 School of Information Engineering, Xinyang Agriculture and Forestry University, Xinyang, Henan, China,
2 College Of Agriculture/ Tree Peony, Henan University Of Science And Technology, Luoyang, Henan, China

* cq@xyafu.edu.cn

## Abstract

Monitoring and precisely localizing rice diseases is essential for agricultural productivity and food security. Existing detection methods face challenges such as high computational complexity, semantic information loss, difficulty detecting small targets, and limited robustness. To address these issues, this study proposes ECL-RTDETR, an enhanced RT-DETR–based rice disease detection model. First, a lightweight Efficient-ViT backbone is employed for feature extraction, incorporating a streamlined multi-head self-attention module to improve inference speed, reduce computational cost, and strengthen local feature extraction. Second, the CARAFE upsampling operator is introduced to better preserve detailed feature information without added computational burden, enhancing fine-grained representation. Finally, standard convolution in the neck network is replaced with LDConv (lightweight dynamic convolution) to enable adaptive feature learning under complex conditions, addressing variations caused by illumination, occlusion, and disease diversity. Experimental results show that ECL-RTDETR improves mAP@0.5 by 0.7%, increases detection speed by 22.2 FPS, and reduces computational cost by 81.8 GFLOPs and parameters by 22.12M compared with the baseline RT-DETR. Overall, ECL-RTDETR delivers superior accuracy, speed, and efficiency, offering a robust solution for intelligent rice disease detection and localization, and advancing smart agriculture and sustainable food security.

## 1. Introduction

Rice diseases pose a major threat to stable and secure rice production. Due to the extensive global cultivation area, significant regional variations, complex climatic conditions, and the biological traits of pests, including large populations, diverse species, and rapid propagation, effective prevention and control remain challenging. The most common rice diseases include bacterial blight, brown spot, and leaf spot [1]. According to statistics, diseases account for over 5% of annual rice yield losses in China [2]. Timely and accurate identification of rice disease types and affected areas at different

**Data availability statement:** Data Availability: All relevant data for this study are publicly available from the figshare repository (https://figshare.com/s/b491aeb44611dea9c481).

**Funding:** This research was supported by the Science and Technology Research Projects of Henan Province (Grant No. 252102111173) awarded to YZ and the Postgraduate Education Reform and Quality Improvement Project of Henan Province (Grant No. YJS2026ZYKC28) awarded to YX. The funders had no role in the design of the study, the collection, analysis and interpretation of data, the writing of the manuscript, or the decision to publish the results.

**Competing interests:** There is no potential conflict of interest in our article, and all authors have seen the manuscript and approved it to submit to your journal.

growth stages is essential for implementing targeted prevention and control strategies. This approach reduces economic losses and minimizes environmental pollution from indiscriminate pesticide use. Traditionally, rice disease monitoring relies on expert observation of symptoms on leaves, stems, and roots through field sampling. However, this subjective and experience-dependent manual identification process is often labor-intensive, time-consuming, and prone to inconsistencies. As a result, it is inadequate for meeting the demands of large-scale, rapid, and accurate disease monitoring in modern agricultural production [3]. Rice diseases pose serious threats to crop growth, directly reducing yield and overall quality. To overcome the limitations of low efficiency and high cost inherent in manual detection methods, and to enable real-time, scalable monitoring, this study focuses on developing precise rice disease recognition technology. The ultimate goal is to achieve targeted disease management, reduce monitoring costs, and enhance rice productivity.

In recent years, the fast progress of deep learning and machine learning technologies has significantly influenced research on rice disease detection. Many researchers have explored the application of diverse deep learning models to identify rice diseases from images. Object detection algorithms based on deep learning are generally classified into two types: one-stage algorithms (e.g., SSD [4], YOLO [5], and RT-DETR [6]) and two-stage algorithms (e.g., R-CNN [7], Fast R-CNN [8], and Faster R-CNN [9]). Zhou et al. [10] introduced a "fast rice disease detection method" that integrates Faster R-CNN and FCM-KM. Their approach first applies "two-dimensional filtering masks" in combination with a "weighted multilevel median filter" (2DFM-AMMF) for noise decline, followed by a "rapid two-dimensional Otsu threshold segmentation algorithm" to minimize background interference during leaf target detection. Experimental findings demonstrated that this method enhanced the Faster R-CNN's recognition accuracy while lowering processing time; however, the model remained large and detection speed was comparatively slow. Shu et al. [11] utilized the SSD algorithm to train a self-constructed RICE dataset, developing a rice panicle detection model suitable for field conditions. Tests revealed a mean average precision (mAP) of 38.1%, indicating limited accuracy and insufficient precision for real-world rice disease identification. Arun et al. [12] applied tiny YOLO (T-YOLO) v4 as a baseline detector and incorporated modules such as spatial pyramid pooling, convolutional block attention module, stacked convolutional feature extraction module, Ghost module, and extra convolutional layers to improve performance. The enhanced drone-based T-YOLO-Rice network achieved an mAP of 86% on a rice leaf disease dataset, yet the accuracy was still inadequate for high-precision detection. Su et al. [13] proposed an improved YOLO-based model (YOLO-DPD) capable of detecting three rice canopy diseases and pests at multiple image scales, achieving an average accuracy of 90.11%, though the model size and inference time remained issues. Hua et al. [14] introduced an enhanced YOLOv5 model integrating the RepVGG network structure, which combines 3×3 convolutions with ReLU and uses a multi-branch topology during training, later optimizing inference speed through layer fusion. Despite these improvements, the model achieved only 90.2% average precision, still below the desired threshold. Chen et al. [15] developed an enhanced

YOLOv6 model by merging 16 visually similar categories into six broader ones, attaining 80.62% accuracy, which was insufficient for accurate rice disease detection. Jia et al. [16] proposed a MobileNet-CA-YOLO model based on YOLOv7, incorporating the lightweight MobileNetV3 network for feature extraction along with coordinate attention (CA) and the SIoU loss function to enhance precision. The model achieved an mAP@0.5 of 93.7%, but remained large and computationally intensive. Dong et al. [17] improved YOLOv8 by integrating EIoU and $\alpha$-IoU loss functions, yielding an 89.9% detection accuracy, still inadequate for precise detection. Wang et al. [18] introduced the RGC-YOLO model, which added a Rep-Ghost structural re-parameterization module to YOLOv8n, resulting in a lightweight model but with only 86.2% accuracy. Similarly, Wang et al. [19] developed the RP-DETR model, extending RT-DETR with a self-designed RepPConv module; however, it achieved only 71.5% accuracy, falling short of practical standards. Song et al. [20] designed an improved RT-DETR-based model, ADAM-DETR, which achieved a detection accuracy of mAP@50 reaching 94.76%, though the model's size and speed limitations persisted.

In summary, current rice disease detection still encounters several significant challenges. First, most existing models are computationally intensive and demand substantial hardware resources for deployment. Second, the semantic information within deep network feature maps often experiences loss, leading to decreased detection accuracy. Finally, detection performance remains weak in scenarios involving complex backgrounds or small targets, indicating that model robustness requires further enhancement.

The key contributions of this work are outlined as follows:

(1) The lightweight EfficientViT network is applied as the backbone for feature extraction. This network attains high computational efficiency via a lightweight multi-head self-attention (MSA) module, enhancing inference speed while sustaining low computational cost and significantly improving local feature extraction through the linear attention mechanism.

(2) The CARAFE upsampling operator is integrated to preserve feature information and maintain computational efficiency, thereby enhancing the detail representation of upsampled feature maps.

(3) The standard convolution (SC) layers in the RT-DETR neck network are replaced with LDConv to adapt to dynamic variations in target features under complex conditions. This modification effectively mitigates feature distortions in rice disease images caused by illumination changes, occlusion, and disease diversity, thereby enhancing localization accuracy and enabling finer feature extraction.

## 2. Improved RT-DETR-based rice disease detection model

### 2.1 RT-DETR model

The RT-DETR [21]CV, developed by Baidu, represents a significant advancement in real-time object detection. As an end-to-end Transformer-based model, RT-DETR decouples intra-scale interactions and cross-scale feature fusion to efficiently process multi-scale features while markedly reducing the computational cost of traditional DETR models. It outperforms YOLO-based models in both speed and accuracy. RT-DETR's architecture includes 3 main components: a backbone network, a feature processing module, and a decoding prediction network. The model structure is demonstrated in Fig 1.

The three primary components of RT-DETR operate with relative independence yet maintain close coordination, collectively forming a highly efficient object detection pipeline. Among these, the backbone network and the feature processing module constitute the model's core structure and directly determine its detection performance. Furthermore, the backbone network extracts "multi-level visual features" from input images, whereas the "feature processing module" handles feature optimization and fusion. Although the conventional RT-DETR utilizes a CNN-based backbone with robust feature extraction competences, it still encounters practical challenges, including a high parameter count and significant computational overhead. In addition, the attention mechanism within the AIFI module adds computational redundancy across

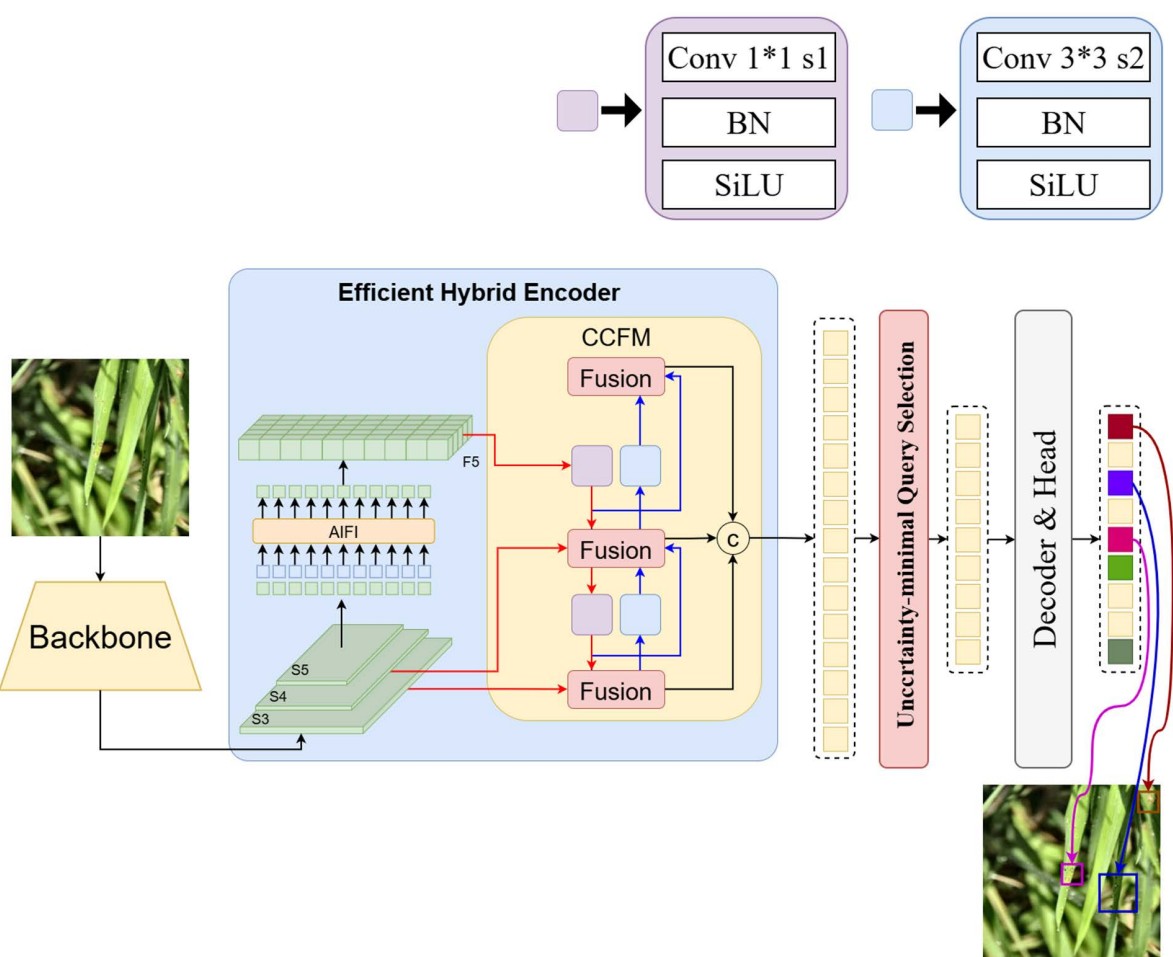

**Fig 1. RT-DETR model diagram.**

various feature levels, further complicating model training. It is also important to note that while the structure of the CCFM module resembles that of the path aggregation feature pyramid network [22], it faces unique challenges in feature integration. In object detection tasks, an ideal feature representation must retain both low-level details and high-level semantic information. However, conventional pyramid structures often experience semantic degradation at higher levels, while bottom-up fusion mechanisms tend to lose essential low-level details, issues that pose momentous problems for small-object detection. Consequently, future research should focus on optimizing backbone networks and feature fusion modules to address these limitations and enhance the detection accuracy of fine-grained targets in agricultural applications such as rice disease detection.

## 2.2 EfficientViT

The design of EfficientViT [23] is inspired by the overall architecture of the ViT (Vision Transformer) but adopts a hybrid modular framework to improve practicality and efficiency. It integrates lightweight convolutional components, such as depthwise separable convolution (DSConv) and MobileNet-style MBConv, alongside the authors' custom EfficientViT modules, achieving an effective balance between computational efficiency and representational power. The complete

architecture of EfficientViT is shown in Fig 2. The network first processes the input image through convolutional (Conv) layers for feature extraction and dimensionality reduction. The output is then refined using the DSConv structure to enhance both accuracy and computational efficiency. Finally, pointwise convolution (MBConv) fuses the depthwise convolution results through a 1×1 kernel to generate the final feature map. Since depthwise convolution operates within each channel independently, this design improves model performance while maintaining high computational efficiency.

The EfficientViT module comprises two main components: a lightweight MSA(Multi-Head Self-Attention) module for contextual feature extraction and an MBConv module for capturing local features. However, the linear attention mechanism in the lightweight MSA module inherently limits fine-grained local detail representation, potentially reducing detection accuracy. To address this limitation, this study incorporates a depthwise convolution–based MBConv module after the MSA module to strengthen the linear attention mechanism. This design enhances local feature extraction while maintaining low computational complexity, thereby improving overall efficiency and accuracy.

The EfficientViT model adopts a "standard backbone–head (decoder) architecture," incorporating several key design characteristics:

(1) The backbone network consists of an input layer followed by four sequential stages, each progressively reducing feature map resolution while increasing the number of channels.

(2) Lightweight multi-input multi-output (MSA) modules are embedded within the third and fourth stages to enhance contextual feature extraction.

(3) For downsampling operations, the model applies MBConv blocks with a stride of 2. The outputs from stages 2, 3, and 4 collectively form a feature pyramid, which serves as input to the neck network for feature fusion.

The detailed architectural configurations of several EfficientViT variants are presented in Table 1.

In this study, the EfficientViT model described above is employed to replace the feature extraction network of RT-DETR, targeting to attain efficient hardware computation through its lightweight MSA design and thereby enhance the model's inference speed. This integration enables the model to attain global perception and multi-scale feature learning without compromising detection accuracy. Consequently, the proposed model is capable of performing real-time rice disease recognition with both high efficiency and precision.

It should be noted that EfficientViT and its related feature enhancement strategies exhibit strong applicability in the scenario of rice disease detection. Rice diseases typically manifest as patch targets with small scales, blurred boundaries, and subtle textural features, and are easily disturbed by complex field backgrounds (e.g., vein structures, soil backgrounds) and illumination variations. By introducing a lightweight MSA (Multi-Head Self-Attention) mechanism, EfficientViT

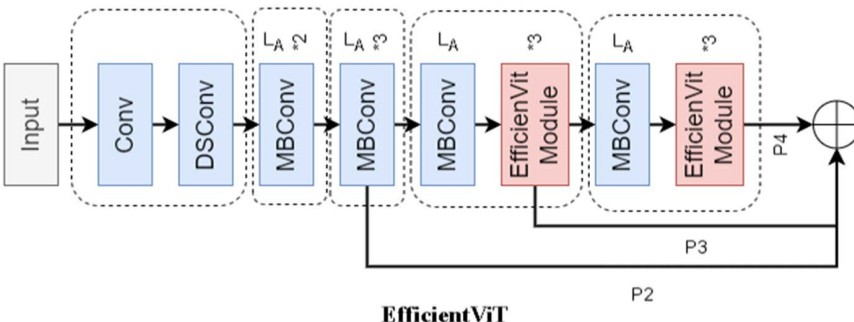

**Fig 2. EfficientViT structure diagram.**

**Table 1. Detailed architectural configurations of different efficientViT variants.**

| Variant | Input | Stage 1 | Stage 2 | Stage 3 | Stage 4 |
|---|---|---|---|---|---|
| C | C0 = 8 | C1 = 16 | C2 = 32 | C3 = 64 | C4 = 128 |
| L | L0 = 1 | L1 = 2 | L2 = 2 | L3 = 2 | L4 = 2 |
| H | 640 | 640 | 640 | 640 | 640 |
| W | 640 | 640 | 640 | 640 | 640 |

Where, C represents the number of channels, L presents the number of blocks, H indicates the feature map height, and W indicates the feature map width.

enhances the modeling capability of global contextual information while maintaining low computational complexity, which helps to distinguish subtle differences between lesion regions and background leaf textures. Furthermore, its integration with convolutional structures empowers the model with superior capacity in capturing local morphological changes of rice diseases (e.g., irregular diffusion characteristics of lesion edges), thereby achieving a favorable balance between accuracy and efficiency in complex agricultural scenarios.

## 2.3 CARAFE upsampling operator

In object detection tasks, the upsampling process can be articulated as the dot product between the upsampling kernel at each position of the feature map and the pixels within its respective receptive region. Conventional upsampling techniques, including nearest-neighbor interpolation and bilinear interpolation, are commonly used due to their simplicity and efficiency. The RT-DETR model employs nearest-neighbor interpolation for this purpose; however, while computationally lightweight, this method fails to fully leverage the semantic information embedded in feature maps, particularly for rice disease detection. Furthermore, its limited receptive field often leads to a loss of fine-grained details. To overcome these limitations, this study introduces the CARAFE [24] upsampling operator to replace the original nearest-neighbor interpolation. As illustrated in Fig 3, the CARAFE operator preserves feature information more effectively while maintaining computational efficiency, thereby enhancing the detailed representation capability of the upsampled feature maps.

In the CARAFE framework, a "feature map of size" $C \times H \times W$, along with an upsampling rate (where $\sigma$ is an integer), is simultaneously fed into both the kernel prediction module and the "content-aware reassembly module." These two modules collaboratively process the input to produce an "output feature map" $x'$ of size $C \times \sigma H \times \sigma W$, thereby achieving the upsampling of the "original feature map." For any given position $l' = (i', j')$ on the output feature map $x'$, there exists a corresponding position $l=(i, j)$ on the "input feature map" $x$, where $i = \lfloor i'/\sigma \rfloor$, $j = \lfloor j'/\sigma \rfloor$. The "kernel prediction module" $\Psi$ adaptively generates a unique reassembly kernel $W_{l'}$ for each target position $l'$ in the output feature map $x'$. The overall computation process can be expressed as follows:

$$W_{l'} = \Psi(N(x_l, k_{encoder}))$$

(1)

Here, $x_l$ denotes a pixel position on the "input feature map" $x$, $k_{encoder}$ represents the reassembly kernel, and $N(x_l, k_{encoder})$ is the target position $l'$ on the input feature map $x$ relative to a square region centered at $l$ with a spatial size of $k_{encoder}$. The kernel prediction module consists of 3 submodules: the channel compressor, the content encoder, and the kernel normalizer. Initially, the channel compressor reduces the dimensionality of the input feature map $C \times H \times W$ into $C_m \times H \times W$. Subsequently, the content encoder applies a convolution operation using a $k_{encoder} \times k_{encoder} \times C_{up}$ kernel to generate a feature map of size $\sigma^2 \times k_{up}^2 \times H \times W$, where $C_{up} = \sigma^2 \times k_{up}^2$ denotes the upsampling rate. Afterward, the resulting feature map $\sigma W \times \sigma H \times k_{up}^2$ is reshaped to match the desired dimensions. Finally, the kernel normalizer applies the softmax function to each spatial position to produce a normalized output kernel, ensuring that the weights sum to one across the reassembly region.

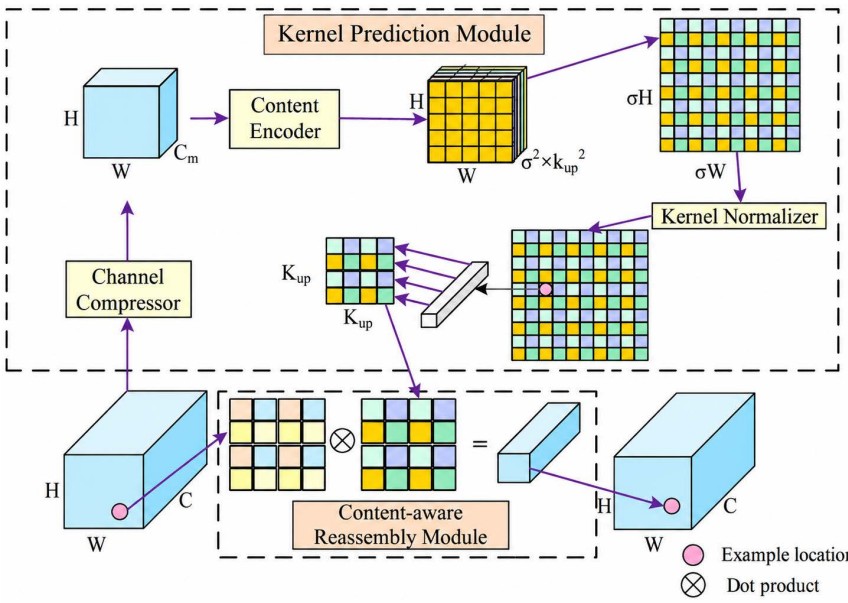

**Fig 3. Structure diagram of the CARAFE module.**

The content-aware reassembly module utilizes a weighted aggregation operation to generate the upsampled feature corresponding to pixel $l'$. This process effectively integrates information from the surrounding neighborhood based on the learned reassembly kernels. The weighting process can be mathematically expressed as follows:

$$x_{l'}' = \phi(N(x_l, k_{up}), W_{l'}) = \sum_{n=-r}^{r} \sum_{m=-r}^{r} W_{l'}(n, m) \cdot x_{(i+n,j+m)}$$

(2)

where $k_{up}$ represents the spatial size of the reassembly kernel, and $N(x_l, k_{up})$ denotes the target position $l'$ on the "input feature map" $x$. This position corresponds to a square region centered at $l$, with dimensions of $k_{up} \times k_{up}$. The term $W_{l'}$ refers to the reassembly kernel that is adaptively generated by the "kernel prediction module" for the specific target position $l'$, $r = \lfloor k_{up}/2 \rfloor$.

In the task of rice disease detection, the introduction of the CARAFE upsampling operator demonstrates clear scene-specific pertinence. Rice disease lesions usually exhibit indistinct edges, irregular morphologies, and significant scale variations. Traditional upsampling methods such as nearest-neighbor or bilinear interpolation tend to cause the loss of lesion boundary information during upsampling, thereby degrading detection performance for small targets and weak texture regions. By dynamically generating reconstruction kernels for different spatial locations in a content-aware manner, CARAFE enables the upsampling process to adaptively focus on lesion regions and their contextual information, thus more effectively preserving lesion edge structures and fine-grained texture features. This characteristic is of great importance for improving the localization accuracy of rice disease targets in complex field environments.

## 2.4 LDConv module

In rice disease detection, target features often exhibit considerable variability due to factors such as changes in illumination, differences in disease manifestation, and occlusion. These variations present significant challenges during the feature fusion stage. SC operations, being static in nature, are unable to adapt to such dynamic changes or adjust sampling

positions in a context-aware manner. This limitation reduces the precision of target localization and increases the likelihood of false detections. Furthermore, the restricted receptive field of SC hampers its ability to efficiently capture fine-grained and discriminative target features. As a result, the loss of crucial detail information further diminishes the model's overall robustness and detection accuracy.

To address the limitations of SC, this study incorporates LDConv [25], a convolutional method designed to manage complex and variable scenarios. The core principle of LDConv is the generation and adaptive utilization of dynamic convolution kernels. Unlike conventional convolution, which applies a "single static kernel" across all spatial positions, LDConv employs multiple parallel static kernels to construct adaptive dynamic kernels. These are dynamically aggregated according to input feature variations, empowering the model to proficiently capture local content differences. The overall architecture of the "LDConv module" is illustrated in Fig 4. Before generating irregular convolution kernels, an initial sampling operation determines the central coordinate ($P_0$) of the kernel, as mathematically defined in Equation (3).

$$Conv\left(P_0\right) = \sum_{P_N \in R} w_{P_N} \times x\left(P_0 + P_N\right)$$

$$(3)$$

In Equation (3), $R$, $w_{P_N}$, and $x(P_0 + P_N)$ respectively denote the coordinate set of the sampling positions covered by the convolution operation, the weights of the convolution units, and the pixel value at position $P_0 + P_N$. Assuming the input feature map has dimensions ($C$, $H$, $W$), the convolution operation produces corresponding offsets with dimensions ($2N$, $H$, $W$). These offsets are added to the original coordinates ($P_N + P_0$) to obtain new sampling coordinates, which define spatially varying sampling patterns across different regions of the feature map. This mechanism allows each convolution kernel to adapt its shape dynamically, capturing structural variations in the input more effectively. Subsequently, the irregularly shaped convolution kernels are then used for interpolation and resampling at updated pixel positions, enabling

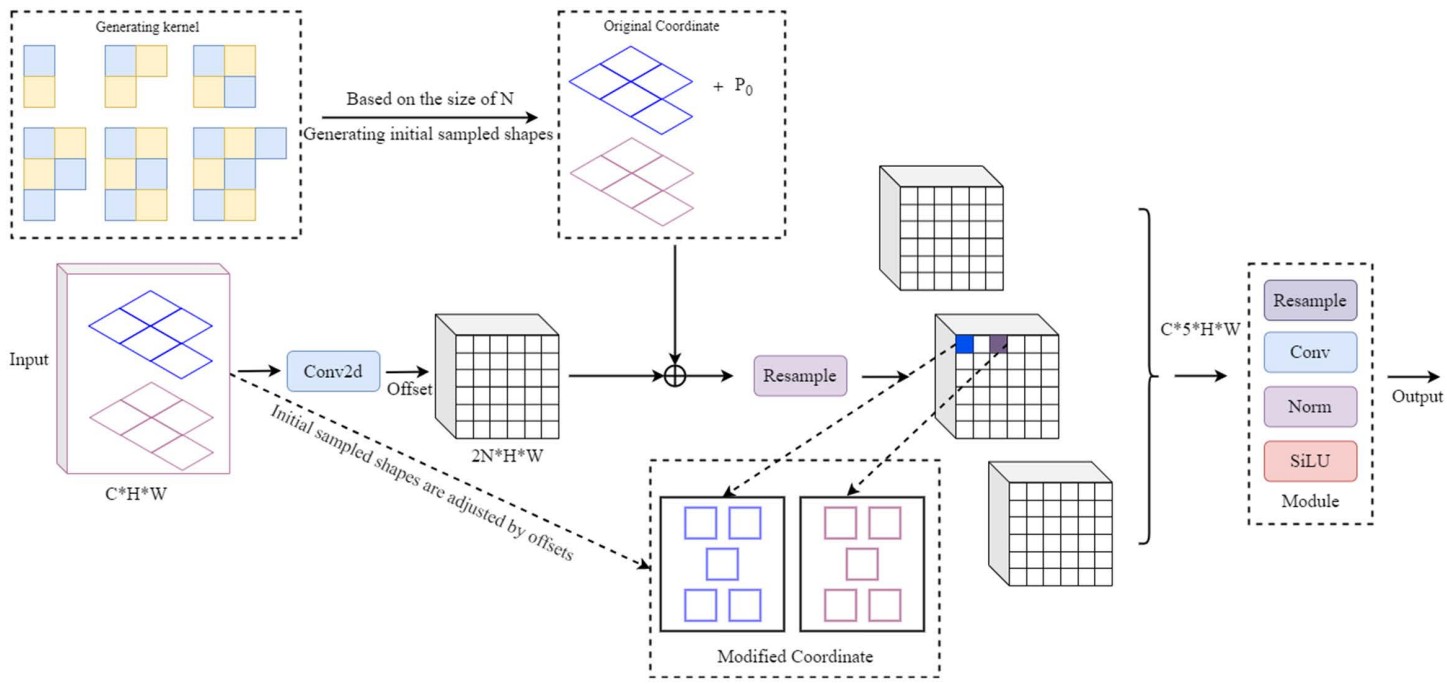

**Fig 4. Structure diagram of the LDConv module.**

the extraction of richer and more fine-grained feature representations. To improve computational efficiency and reduce memory access, convolution, the resampling, and normalization processes are integrated into a unified submodule within the LDConv framework, minimizing intermediate storage and optimizing the overall processing pipeline.

In the experiments of this paper, the specific parameter settings of the LDConv module are as follows: the convolution kernel size is set to 3×3 with a stride of 1, and the number of sampling points is consistent with that of standard convolution; the dynamic offset branch is predicted using a lightweight convolutional structure, whose channel number is identical to the input feature channels, and the extra parameters are reduced by weight sharing. Under the above configuration, LDConv introduces only a small amount of additional parameters and computational overhead compared with standard convolution, and its impact on the overall model parameters and FLOPs has been verified in the subsequent ablation experiments and overall performance comparison.

In addition, the LDConv module also exhibits obvious advantages in the scenario of rice disease detection. In actual field environments, rice leaves often suffer from occlusion, overlapping and attitude variations, and disease lesions show significant morphological differences at different growth stages and disease severity levels. Due to the fixed sampling positions, traditional standard convolution is difficult to adapt to these dynamic changes, which easily leads to inaccurate target localization. By introducing a dynamic sampling offset mechanism, LDConv enables the convolution kernel to adaptively adjust the sampling positions according to the input features, thereby more flexibly capturing the key features of disease lesions under complex backgrounds and irregular morphologies. Such dynamic modeling capability helps to improve the robustness and generalization ability of the model for rice disease detection in complex agricultural scenarios.

### 2.5 Improved model proposed in this paper: ECL-RTDETR

To address the limitations of the RT-DETR model, such as large model size, frequent false detections of small targets, and limited adaptability to dynamic feature variations, this study introduces targeted enhancements to its backbone, upsampling operator, and neck network. The original backbone is replaced with the lightweight EfficientViT network, which incorporates a streamlined MSA module for efficient hardware computation and faster inference. This modification strengthens the linear attention mechanism's capacity for local feature extraction while maintaining low computational cost. The CARAFE upsampling operator is then integrated to preserve richer feature information and finer detail representation with high computational efficiency. Finally, SC layers in the neck network are replaced with LDConv (dynamic convolution) layers, enabling adaptive feature extraction in complex environments. LDConv achieves this by generating dynamic kernels through parallel static convolutions and aggregating them according to input feature characteristics. This improvement effectively mitigates the challenges posed by illumination changes, occlusions, and disease-induced variations in rice disease images, resulting in enhanced localization precision and finer feature extraction, particularly for small targets and cluttered backgrounds. The combined introduction of the EfficientViT backbone, CARAFE operator, and LDConv dynamic convolution constitutes the optimized model, termed EfficientViT-CARAFE-LDConv-RTDETR (ECL-RTDETR), whose overall architectural framework is illustrated in Fig 5.

## 3. Experiments and results analysis

### 3.1 Data image collection

The rice disease images in this dataset were collected in Minggang Town, Pingqiao District, Xinyang City, between mid-July and the end of September 2024, using a DJI T25P drone with a resolution of 1920×1080 pixels. To account for the impact of natural lighting on image quality, data acquisition was conducted during two daily time windows: 9:00 a.m. to 12:00 noon and 2:00 p.m. to 5:00 p.m., with all images captured from an overhead perspective. To enhance sample diversity and improve model robustness, images were collected under both sunny and cloudy conditions at different times of the day, as illustrated in Fig 6. Initially, 1,988 images were captured on sunny days and 2,069 on cloudy days. After careful screening for image quality and optimal shooting angles, a total of 2,579 high-quality original images were retained. The

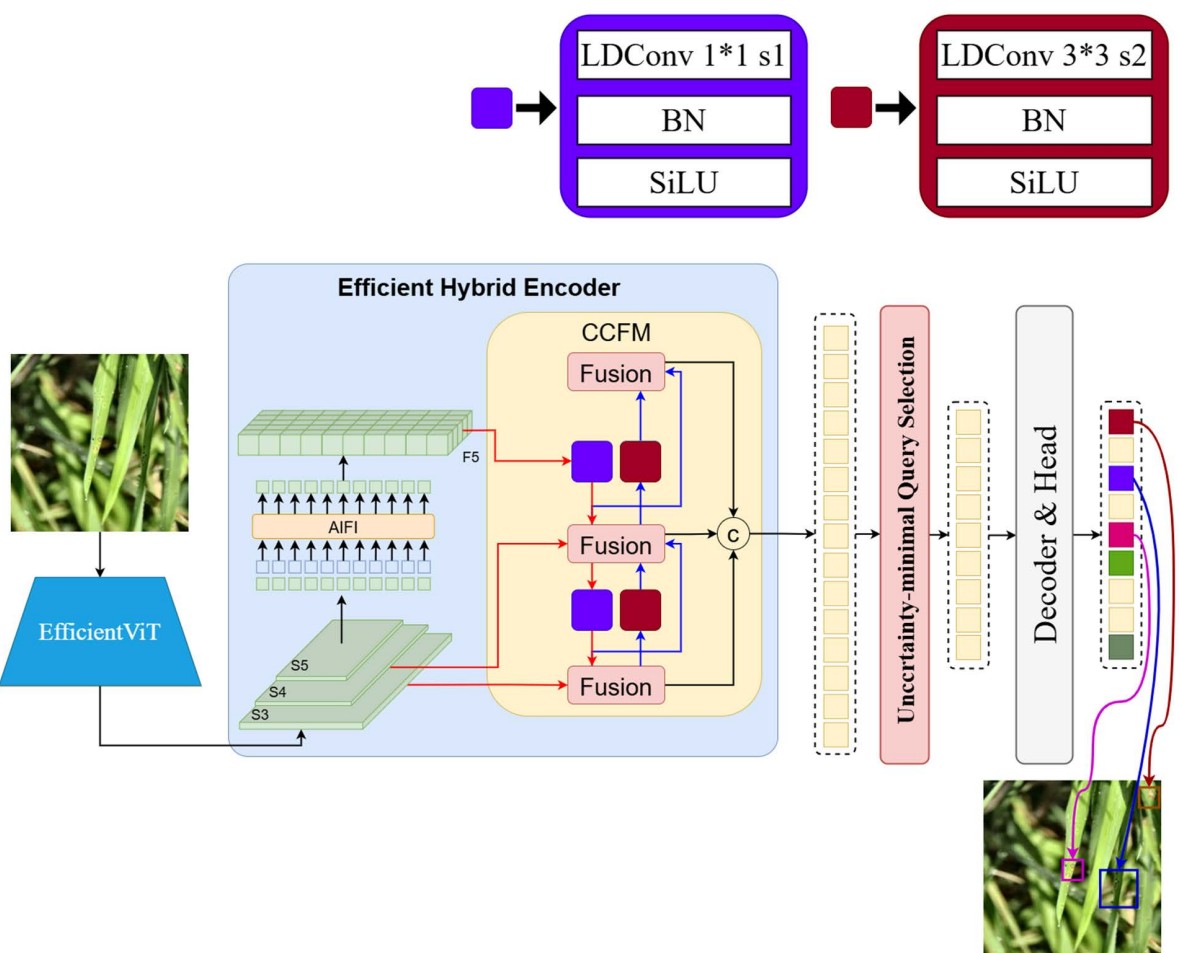

**Fig 5. ECL-RTDETR model diagram.**

dataset encompasses three prevalent rice diseases: brown spot, bacterial blight, and leaf spot, with their corresponding labels and counts detailed in Table 2. Example images from the dataset are presented in Fig 7.

### 3.2 Dataset preprocessing

**3.2.1 Image annotation.** In this study, LabelImg [26] (version 1.8.1) was employed to manually annotate the positions of rice diseases in each image. LabelImg is a widely used image annotation tool designed for marking objects of interest, particularly in the preparation of datasets for object detection tasks. It enables users to create rectangular bounding boxes around target objects using a straightforward graphical interface and assign a corresponding category label to each box. The LabelImg interface is illustrated in Fig 8. The annotation process generates XML files that store the coordinates of each bounding box. Converting these coordinates into ratios relative to image resolution transforms the data into YOLO's required label file format, enabling seamless integration into object detection pipelines.

**3.2.2 Dataset augmentation.** To enhance generalization and reduce overfitting from limited training data, various data augmentation techniques were applied to expand the original image set. As shown in Fig 9, these include noise addition,

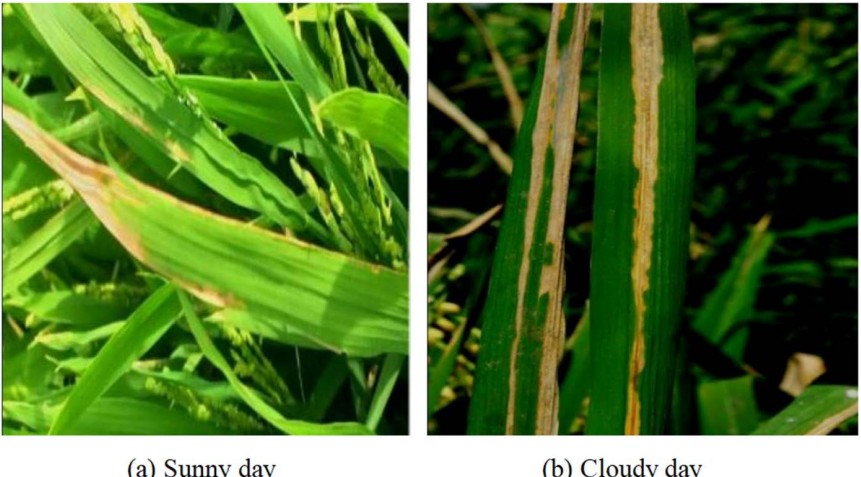

(a) Sunny day          (b) Cloudy day

**Fig 6. Sample images under sunny and cloudy conditions.**

**Table 2. Correspondence of rice disease labels.**

| Category | Corresponding Label | Number of Images |
|---|---|---|
| Bacterial Blight | Bacteria_Leaf_Blight | 589 |
| Brown Spot | Brown_Spot | 1137 |
| Leaf Smut | Leaf_Smut | 853 |

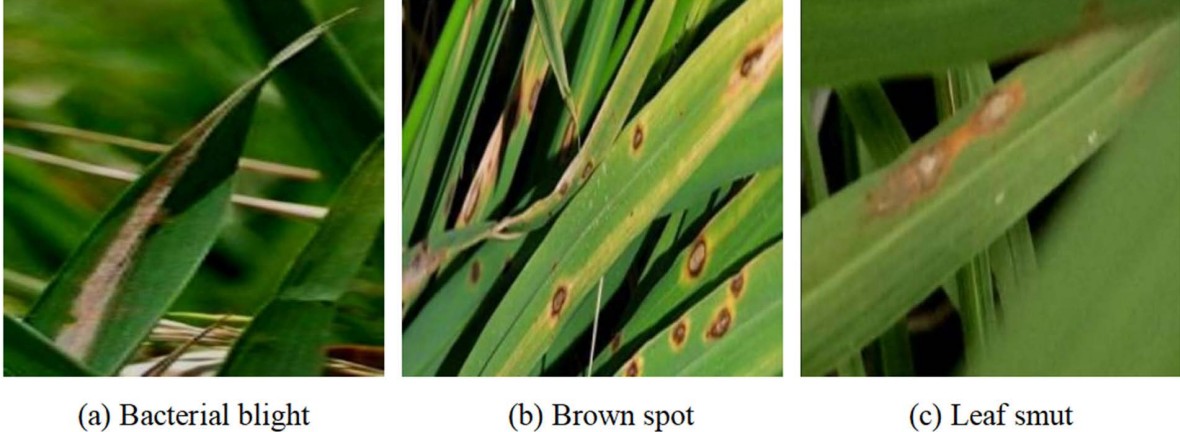

(a) Bacterial blight          (b) Brown spot          (c) Leaf smut

**Fig 7. Sample Images of Rice Disease Data.**

rotation, random brightness adjustment, mirroring, and grayscale conversion. After augmentation, the dataset comprised 6,204 images, split in a 7:2:1 ratio for training, validation, and testing. These strategies simulate real-world imaging variability, improving the model's adaptability to diverse scenarios.

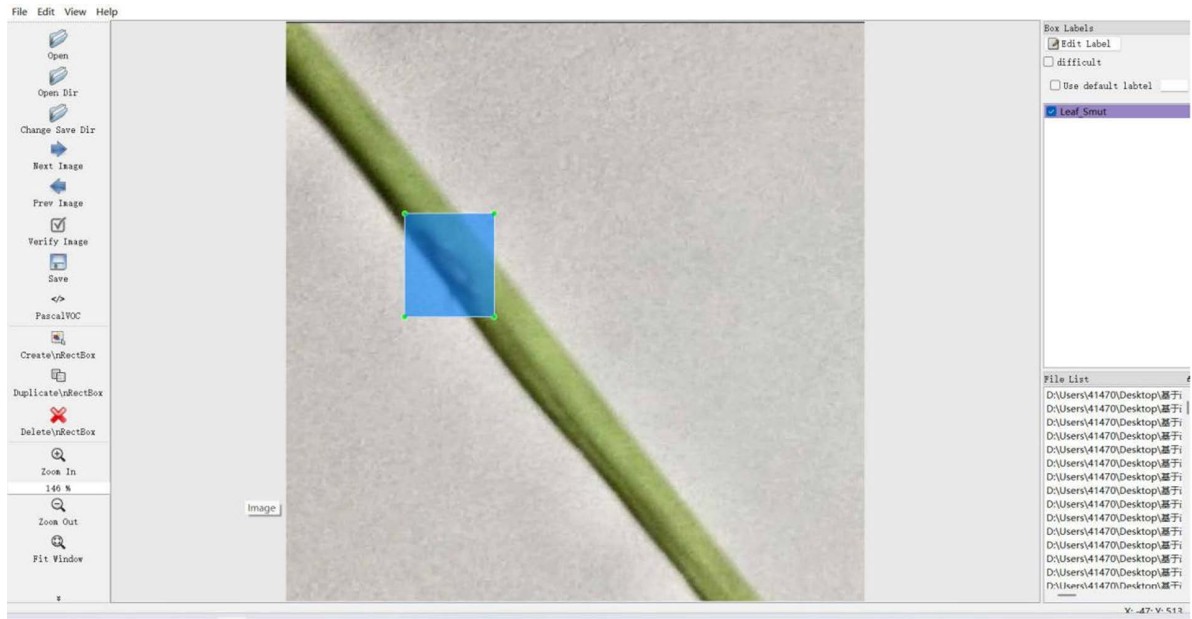

**Fig 8. Labellmg annotation interface.**

### 3.3 Experimental environment

To evaluate ECL-RTDETR, experiments were conducted on an Ubuntu 20.04 system using PyTorch 1.10.0. The original RT-DETR model served as the baseline. Hardware and software configurations are summarized in Table 3.

   All experiments used consistent hyperparameters, detailed in Table 4. Input images were resized to 640×640 pixels, and the Adam optimizer was employed with a momentum factor of 0.937. Key settings included an initial learning rate of 0.01, batch size of 16, 200 training epochs, and weight decay of 0.0005. These configurations ensured experimental stability and reliable rice disease detection under complex environmental conditions.

   In addition, the YOLOv11n and YOLOv12n models used for comparison in this study were implemented based on the official Ultralytics GitHub repository (https://github.com/ultralytics/ultralytics.git). All models retained their default network architectures and were trained and evaluated under a unified hardware–software environment with consistent training parameters, ensuring the reproducibility of experimental results and the fairness of comparisons.

### 3.4 Evaluation metrics

This study aims to provide a comprehensive evaluation of model performance using multiple metrics, including Precision (P), Recall (R), mAP, Giga floating point operations per second (GFLOP), number of parameters, and detection speed. The formulas for calculating P, R, and mAP are presented in Equations (4)–(6).

$$P = \frac{T_P}{T_P + F_P} \times 100\%$$

(4)

$$R = \frac{T_P}{T_P + F_N} \times 100\%$$

(5)

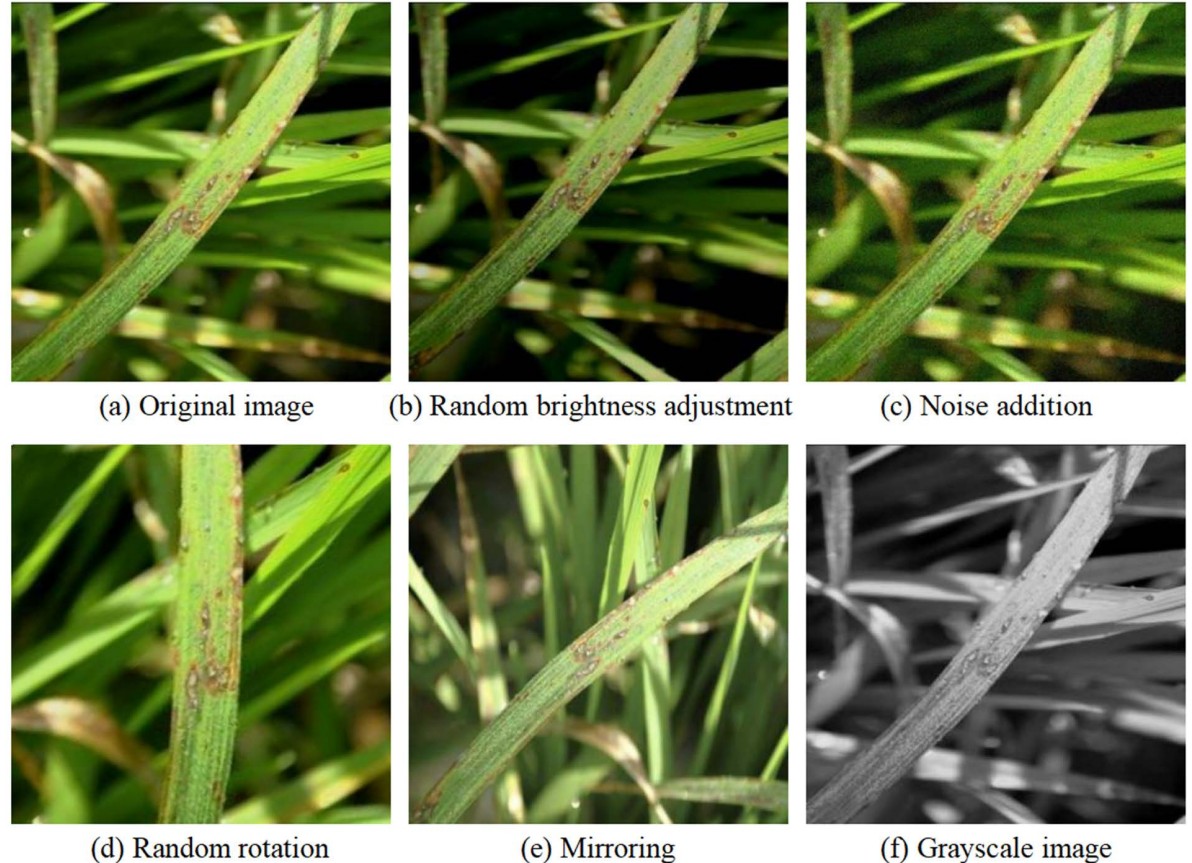

(a) Original image  (b) Random brightness adjustment  (c) Noise addition

(d) Random rotation  (e) Mirroring  (f) Grayscale image

**Fig 9. Examples of data augmentation.**

**Table 3. Experimental environment configurations.**

| Hardware configuration | | Software configuration | |
|---|---|---|---|
| GRAPHICS CARD | RTX 3090 | System | Ubuntu20.04 |
| HARD DISK | SSD NVMe 1TB | Python Version | 3.8.12 |
| MEMORY | 16GB RAM | Framework | Pytorch 1.10 + cudnn8.0.5 |
| PROCESSOR | Intel i5-1035G1 CPU | CUDA | CUDA11.1 |

$$FPS = \frac{N}{t}$$
(6)

Here, $T_P$ represents the number of correctly identified true positive instances, $F_P$ denotes the number of false positive instances, and $F_N$ indicates the number of false negative instances. Where, $N$ represents the number of images to be detected, and $t$ represents the detection time.

## 3.5 Experimental results and analysis

### 3.5.1 Comparative experiments on backbone networks.
It should be noted that the experiments in this section are mainly conducted to analyze the influence of different backbone network structures on the overall performance of the

**Table 4. Training parameters.**

| Category | Setting value |
| --- | --- |
| INPUT SIZE | 640*640 |
| Optimizer | Adam |
| Momentum Factor (momentum) | 0.937 |
| Optimizer Weight Decay (weight_decay) | 0.0005 |
| Learning Rate (learning_rate) | 0.01 |
| Number of Epochs (epoch) | 200 |
| Batch Size (batch_size) | 16 |
| Subdivisions | 4 |
| Pretrained Weights | rtdetr-l.pt |
| Training Set: Validation Set: Test Set | 7: 2: 1 |

RT-DETR model, with a focus on the differences of various backbones in terms of detection accuracy, inference speed and computational complexity, rather than an ablation analysis for each improved module individually.

In rice cultivation and crop management, accurate identification of rice diseases is a highly challenging task. This complexity arises because diseases often manifest as small spots or subtle texture variations in images. Moreover, complex backgrounds and significant illumination changes can easily interfere with accurate recognition. To support lightweight model deployment, this study optimized the baseline model's backbone by replacing the original structure with the lightweight EfficientViT network. Comparative experiments across various conventional backbone networks (see Table 5) demonstrated the superior performance of EfficientViT in recognizing rice diseases.

In this ablation study, the performance of various backbone networks and optimization modules for rice disease detection was compared. The results indicate that RTDETR-R34 and RTDETR-R50, based on the traditional ResNet family, exhibited strong performance in both accuracy and inference speed, with RTDETR-R50 achieving an mAP of 93.5% and an FPS of 19.7, representing a well-balanced trade-off. However, increasing network depth with RTDETR-R101 (ResNet-101) further improved mAP to 95.3%, but inference speed decreased sharply to 3.6 FPS, and the model's complexity and parameter count increased substantially, imposing a heavy computational burden. The main reason lies in the deeper network architecture of ResNet-101, which introduces more convolutional operations and feature maps during feature extraction. This significantly increases computational overhead (with FLOPs reaching up to 257.7 G) and memory access costs, ultimately leading to an obvious decline in inference speed. Introducing modern architectures such as ConvNeXtV2 (RTDETR-Co) and CSWinTransformer (RTDETR-CS) further enhanced accuracy to 95.9% and 96.0%, respectively. Nevertheless, the inference speed is relatively slow (9.5 and 9.9 FPS, respectively). This is because such models typically adopt more sophisticated feature extraction architectures, such as large convolutional kernels, hierarchical designs, or self-attention mechanisms. Although these structures enhance feature representation capability, they also introduce higher computational complexity and inference latency. This indicates that although these networks provide stronger feature representation, they still pose computational challenges. In contrast, the lightweight RTDETR-L network achieved 96.8% accuracy with only 32.9 million parameters and a model size of 63.0 MB, while maintaining an increased FPS of 10.6. This demonstrates an excellent balance between accuracy and efficiency, making RTDETR-L highly suitable for deployment in real-world scenarios with limited computational resources. Finally, although RTDETR-E (EfficientViT) achieved a slightly lower accuracy of 96.6%, it demonstrated excellent inference speed at 31.6 FPS. Its low computational cost and reduced parameter count make it well-suited for real-time inference in resource-constrained environments. Overall, RTDETR-E offers an optimal balance between accuracy and efficiency, making it ideal for edge computing applications such as mobile and embedded device deployment.

**Table 5. Performance comparison of ablation experiments on optimized modules.**

| Model | Backbone Network | mAP 0.5/% | Size (MB) | Parameters/M | FLOPs/G | FPS (f/s) |
|---|---|---|---|---|---|---|
| RTDETR-R34 | ResNet-34 | 90.3% | 17.8 | 90.6 | 31.4 | 49.8 |
| RTDETR-R50 | ResNet-50 | 93.5% | 19.7 | 98.3 | 36.6 | 50.3 |
| RTDETR-R101 | ResNet-101 | 95.3% | 3.6 | 257.7 | 76.6 | 148.9 |
| RTDETR-Co | Convnextv2 | 95.9% | 9.5 | 112.4 | 34.8 | 65.8 |
| RTDETR-CS | CSWinTransformer | 96.0% | 9.9 | 119.6 | 29.3 | 67.7 |
| RTDETR-L | – | 96.8% | 10.6 | 108.3 | 32.9 | 63.0 |
| RTDETR-E | EfficientViT | 96.6% | 31.6 | 28.7 | 11.0 | 42.5 |

**3.5.2 Ablation experiments on optimization modules.** Based on the comparative experiments of different backbone networks (Table 5), to further verify the specific contributions of each optimization module to model performance, this paper conducts systematic ablation experiments on EfficientViT, the CARAFE upsampling operator, and the LDConv dynamic convolution module. By gradually incorporating different modules, we quantitatively analyze their impacts on detection accuracy, inference speed, the number of parameters, and computational complexity. The corresponding experimental results are presented in Table 6. Four groups of experiments are designed to evaluate the different improvements. Each group adopts identical training parameters but is tested on different model configurations. Where "✓" indicates that a particular strategy was applied in the improved model, and "×" denotes its absence. This setup enables the evaluation of each module's contribution to overall network performance and the determination of their relative importance. Comparative analysis across models quantifies each module's impact, offering practical guidance for further optimization.

The ablation experiments clearly demonstrate the impact of each optimization module on RT-DETR performance. In this study, the EfficientViT, CARAFE, and LDConv modules were sequentially integrated to assess their individual contributions and guide further optimization. The baseline RT-DETR model achieved a high mAP of 96.8% but showed limitations in inference speed, FLOPs, and parameter count. Incorporating EfficientViT slightly reduced mAP to 96.6%, while significantly increasing FPS to 31.6, indicating improved inference efficiency and reduced computational demand. Adding CARAFE (RTDETR-EC) raised mAP to 97.6%, with a minor decrease in FPS, suggesting enhanced accuracy at a slight cost to speed. Finally, integrating all modules yielded the ECL-RTDETR model, which maintained excellent overall performance, achieving balanced optimization in accuracy, speed, and resource efficiency. Although mAP slightly decreased to 97.5%, the model achieved a balanced trade-off among accuracy, speed, and resource usage. Overall, the proposed ECL-RTDETR represents the optimal configuration, offering substantial improvements across performance metrics compared with the original RT-DETR and achieving an effective balance between accuracy, efficiency, and computational cost.

By integrating these three optimized modules, ECL-RTDETR demonstrates significant advantages over the original RT-DETR. Fig 10 visually compares the performance differences among models, clearly illustrating these improvements.

From Fig 10(a), it is evident that after 50 training epochs, the loss curves of all models fluctuate within a relatively narrow range. Notably, the ECL-RTDETR model consistently exhibits the lowest loss, indicating smaller discrepancies between its predictions and the actual labels on the training data. This suggests superior generalization ability for real-world applications. As shown in Fig 10(b), the mAP50 curves of the ECL-RTDETR and RTDETR-EC models are relatively close, both consistently outperforming the RTDETR-E and the original RT-DETR models, with faster convergence speeds. Fig 10(c) further confirms that the ECL-RTDETR model maintains a leading position in the mAP50–95 metric, achieving the highest performance across the entire training process. These results indicate that the ECL-RTDETR model excels across all evaluation metrics, combining fast convergence with high accuracy, thereby validating its superiority in rice disease detection. Furthermore, the model's outstanding performance demonstrates its enhanced generalization capability,

**Table 6. Performance comparison of ablation experiments on optimized modules.**

| Model | EfficientViT | CARAFE | LDConv | P/% | R/% | mAP 0.5/% | FPS (f/s) | FLOPs/G | Parameters/M | Size (MB) |
|---|---|---|---|---|---|---|---|---|---|---|
| RT-DETR | × | × | × | 97.3% | 96.7% | 96.8% | 10.6 | 108.3 | 32.9 | 63.0 |
| RTDETR-E | ✓ | × | × | 97.0% | 96.2% | 96.6% | 31.6 | 28.7 | 11.01 | 42.5 |
| RTDETR-EC | ✓ | ✓ | × | 98.1% | 97.0% | 97.6% | 30.2 | 28.6 | 11.05 | 43.1 |
| ECL-RTDETR | ✓ | ✓ | ✓ | 98.3% | 96.7% | 97.5% | 32.8 | 26.5 | 10.78 | 41.3 |

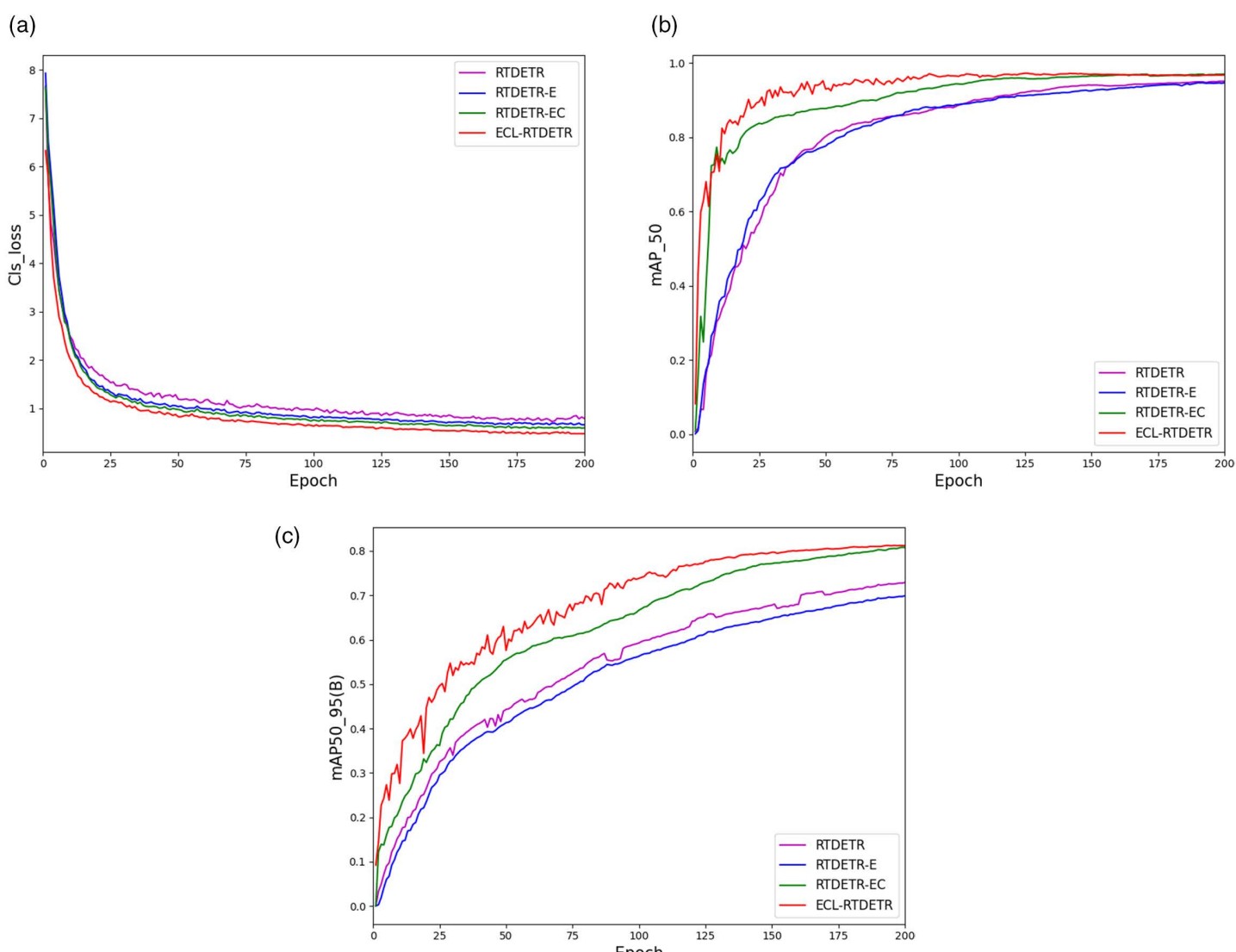

**Fig 10. Comparison charts of different metrics.**

enabling adaptation to more diverse datasets and real-world conditions, which provides significant practical value for rice disease management.

**3.5.3 Comparative experiments.** To further evaluate the detection accuracy and speed of the proposed model, this study conducted a comparison across eight representative network models: Faster R-CNN, SSD, YOLOv5n, YOLOv8n, YOLOv11n, YOLOv12n, RTDETR-L, and ECL-RTDETR. All models were trained using the same dataset. Rather than fixing the number of training epochs, training was dynamically terminated based on each model's actual convergence behavior to ensure that every model achieved its optimal performance. The experimental results are summarized in Table 7.

Table 7 provides a performance comparison between the proposed ECL-RTDETR model and several state-of-the-art object detection methods. Compared with Faster R-CNN, ECL-RTDETR demonstrates superior accuracy, faster inference speed, and greater computational efficiency. The SSD model, however, exhibits relatively low accuracy and fails to meet precision requirements. In comparison with the YOLO series, ECL-RTDETR shows a clear advantage in overall accuracy. Specifically, YOLOv5n achieves an mAP of 88.0%, YOLOv8n reaches 90.6%, YOLOv12n achieves 92.7%, YOLOv11n reaches 93.2%, RTDETR-L achieves 96.8%, while ECL-RTDETR attains 97.5%. Accordingly, the ECL-RTDETR model improves accuracy by 9.5%, 6.9%, 4.8%, 4.3%, and 0.7% over these respective models.

In addition, ECL-RTDETR exhibits notable advantages over other models in terms of parameter count (10.78M), GFLOPs (26.5G), and inference speed (32.8 FPS). Compared with RTDETR-L, ECL-RTDETR maintains high detection accuracy and fast inference, while significantly reducing computational cost, achieving a well-balanced trade-off between performance and efficiency.

To provide a clearer visualization of model performance across multiple metrics, the data from Table 7 were plotted using a radar chart, as shown in Fig 11. In this chart, each model's performance is represented by a curve: the closer the curve is to an axis, the higher the score on the corresponding metric; simultaneously, a larger area enclosed by the curve indicates stronger overall performance across all evaluated metrics.

ECL-RTDETR demonstrates the best overall performance, particularly excelling in mAP, precision, and recall. Although its FPS and model size are slightly lower than those of the YOLO series, its substantially higher mAP indicates superior detection accuracy. In comparison, YOLOv11n and other YOLO models achieve higher FPS and greater efficiency regarding computational complexity and model size. RTDETR-L performs well in precision and recall but is hindered by higher computational demands and lower efficiency relative to the YOLO series. Both SSD and Faster R-CNN exhibit weaker overall performance, particularly in precision and recall, where they are clearly outperformed by the YOLO and RT-DETR models. Consequently, ECL-RTDETR provides significant advantages for practical rice disease detection, offering enhanced applicability and adaptability in real-world scenarios.

**Table 7. Performance comparison of mainstream models.**

| Model | P/% | R/% | mAP0.5/% | FPS(f/s) | FLOPs/G | Parameters/M | Size(MB) |
|---|---|---|---|---|---|---|---|
| FASTER R-CNN | 80.9% | 78.2% | 79.4% | 5.9 | 205 | 63.5 | 164 |
| SSD | 76.4% | 72.9% | 74.2% | 27.1 | 29.3 | 25 | 90.3 |
| YOLOv5n | 89.1% | 86.5% | 88.0% | 36.7 | 6.4 | 3.415 | 14.9 |
| YOLOv8n | 91.4% | 88.2% | 90.6% | 41.9 | 8.9 | 3.151 | 6.24 |
| YOLOv11n | 94.5% | 92.1% | 93.2% | 43.3 | 6.5 | 2.616 | 5.23 |
| YOLOv12n | 94.1% | 91.7% | 92.7% | 40.3 | 6.9 | 4.213 | 6.03 |
| RTDETR-L | 97.3% | 96.7% | 96.8% | 10.6 | 108.3 | 32.9 | 63.0 |
| ECL-RTDETR | 98.3% | 96.7% | 97.5% | 32.8 | 26.5 | 10.78 | 41.3 |

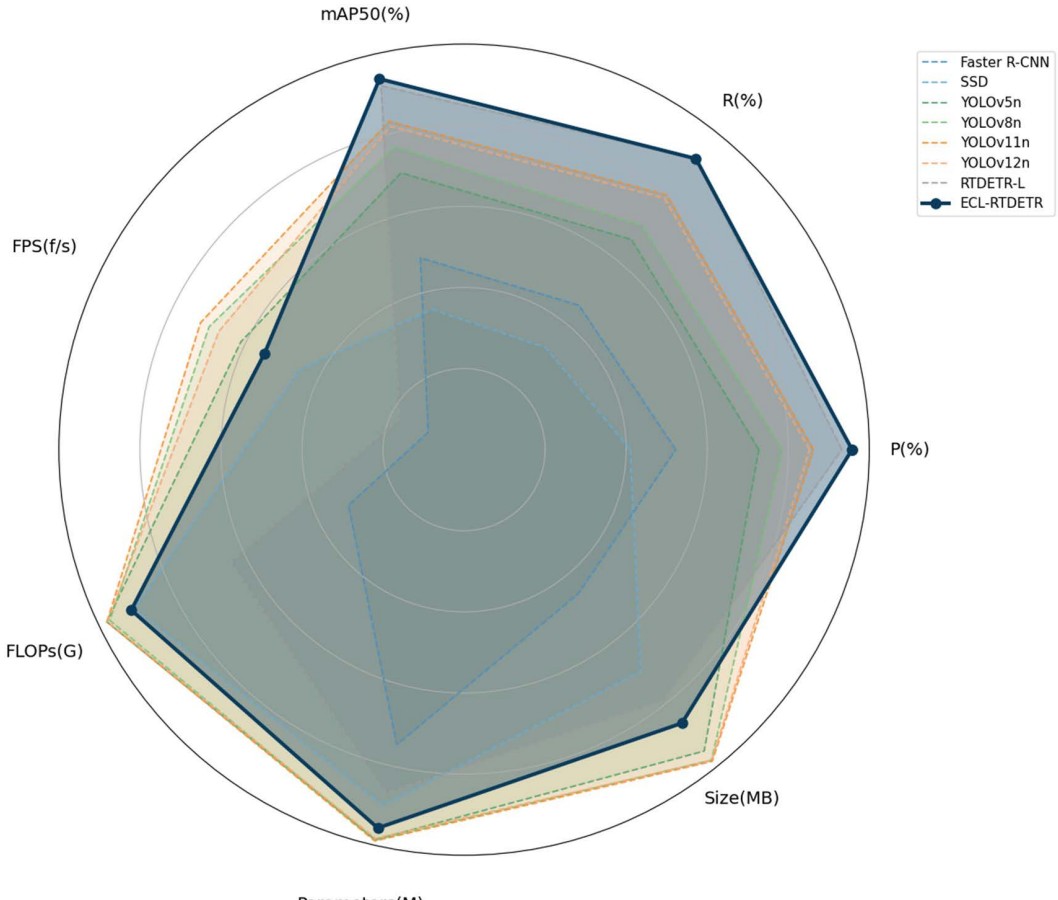

**Fig 11. Radar chart of multi-model performance comparison.**

### 3.6 Comparison of detection results

To comprehensively evaluate the detection performance of the improved algorithm proposed in this paper, a visual analysis was performed on the test set of the self-built dataset, as shown in Fig 12. Fig 12 illustrates the detection results of several mainstream models, including YOLOv8n, YOLOv11n, YOLOv12n, RT-DETR, and ECL-RTDETR.

From the overall detection results, it can be observed that all models can identify rice disease targets to a certain extent, but there are obvious differences in detection completeness, localization accuracy, and adaptability to complex scenes. YOLOv8n, YOLOv11n, and YOLOv12n achieve relatively accurate detection when the targets are clear and of moderate scale. However, they still suffer from varying degrees of missed detection or localization deviation in regions with small-scale targets or strong background interference. The RT-DETR model exhibits problems of target omission and local duplicate detection in complex backgrounds or dense target scenes, and its detection stability needs to be further improved.

In contrast, the ECL-RTDETR model achieves superior detection performance in various scenarios. It obtains more complete and accurate detection results in small-target recognition, dense-target discrimination, and localization accuracy under complex backgrounds. The detection boxes are more consistent with the real disease regions, the target boundaries are more precise, and the overall detection results are more stable.

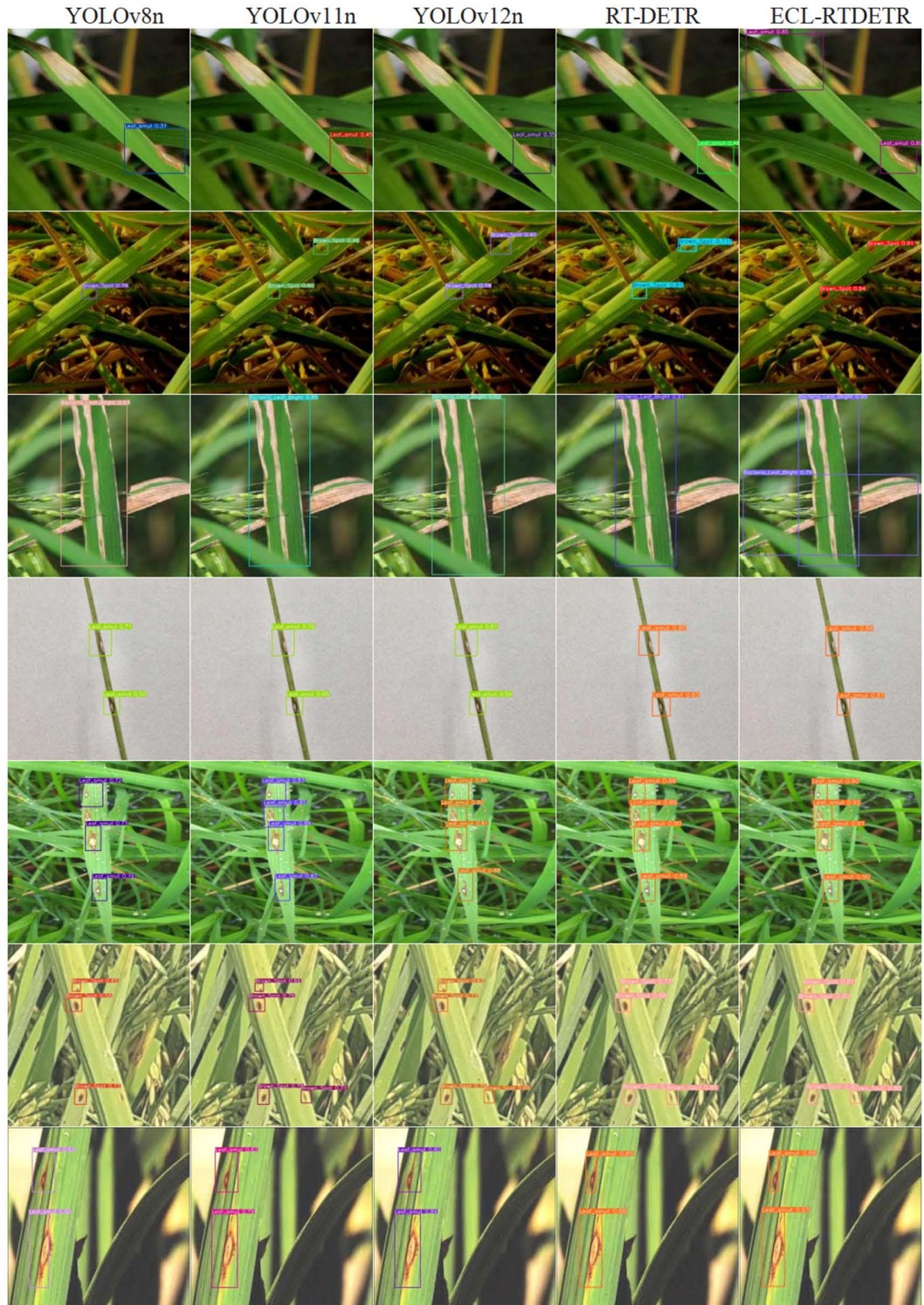

**Fig 12. Comparison of detection performance of different models on the test set.**

Comprehensive analysis of the visualization results shows that the ECL-RTDETR model proposed in this paper outperforms the comparison models in terms of target completeness, localization accuracy, and robustness to complex scenes, which further verifies the effectiveness of the improved strategies in enhancing rice disease detection performance.

To intuitively demonstrate the effectiveness of different models in learning rice disease features, this study adopts the Grad-CAM [27] visualization technique to generate heatmaps for the improved ECL-RTDETR model and the original RT-DETR model, as shown in Figs 13–14. To enhance the sufficiency and representativeness of the visualization results, multiple groups of samples with different disease types, illumination conditions, and background complexities are selected for comparative analysis. The core idea of Grad-CAM is to utilize the feature maps from the last convolutional layer of the deep convolutional neural network (CNN) and calculate the gradient-based importance of different regions. By computing the gradient-weighted activation values for a specific category, Grad-CAM can identify the neurons that contribute most to the model's decision-making process. In the generated heatmaps, red regions represent areas with significant contributions to the prediction, while blue regions have relatively weak influence. This visualization helps to analyze the attention mechanism of the model and understand its decision basis.

Heatmaps in Figs 13–14 reveal that the original RT-DETR model displays dispersed attention during rice disease detection, with limited focus on target regions. In contrast, the proposed ECL-RTDETR model exhibits more concentrated attention, effectively suppressing irrelevant areas and reducing missed detections. These results confirm that ECL-RTDETR enhances target focus and detection accuracy, validating the effectiveness of the proposed improvements.

## 4. Conclusion

Rice diseases significantly affect both yield and quality, underscoring the need for automatic and accurate detection systems. To address the limitations of the RT-DETR model, such as high computational cost, semantic information loss,

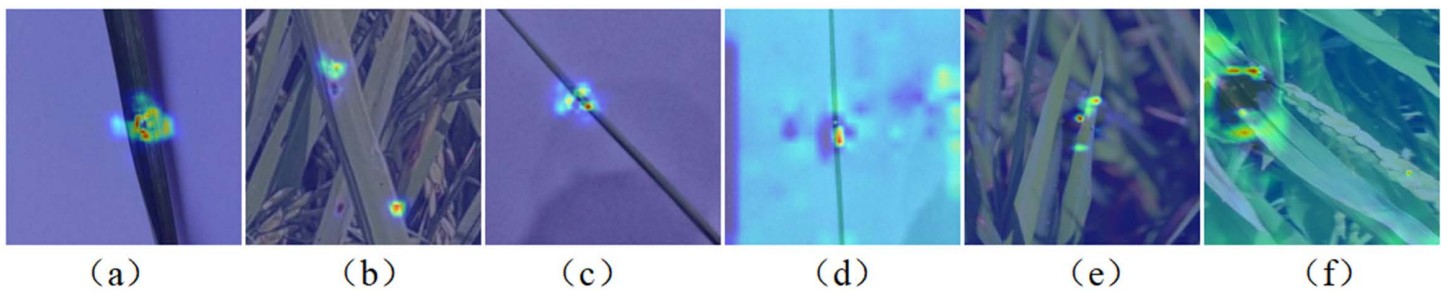

**Fig 13. Grad-CAM heatmaps of RT-DETR.**

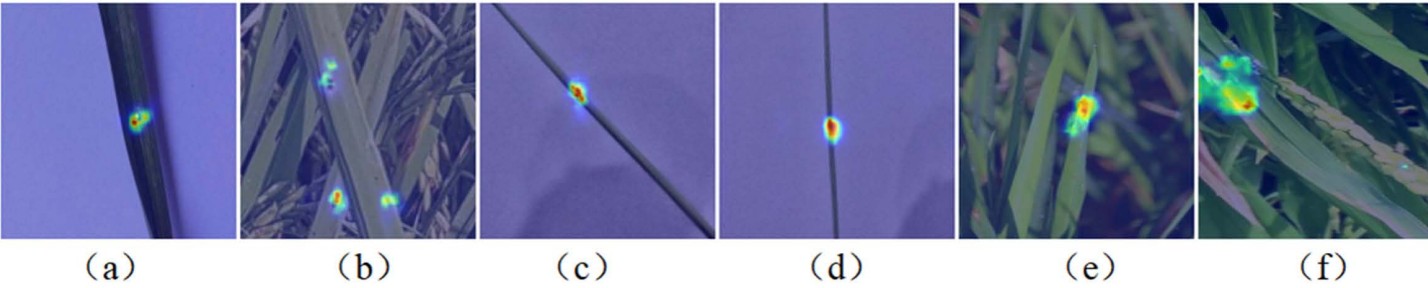

**Fig 14. Grad-CAM heatmap of ECL-RTDETR.**

poor small-target detection, and limited robustness, this study introduces an enhanced RT-DETR-based model named ECL-RTDETR.

The model integrates the lightweight EfficientViT network as its backbone for feature extraction, utilizing a simplified MSA module to optimize hardware computation and improve inference efficiency while maintaining low computational demand. This enhancement strengthens the linear attention mechanism's ability to capture local features. For upsampling, the CARAFE operator replaces traditional nearest-neighbor interpolation, preserving more feature details and improving feature map representation with minimal computational overhead. In the neck network, LDConv (dynamic convolution) substitutes SC to adapt to target variations under complex conditions. This modification mitigates feature distortion caused by illumination changes, occlusion, and disease diversity, thereby enhancing localization precision and fine-grained feature extraction.

Experimental results demonstrate that, on the rice disease dataset, ECL-RTDETR improved mAP@0.5 by 0.7% compared with the original RT-DETR model. Moreover, detection speed increased by 22.2 FPS, while FLOPs and parameters were reduced by 81.8G and 22.12M, respectively. Overall, ECL-RTDETR achieved higher accuracy, faster inference, and fewer missed detections, confirming its effectiveness for practical rice disease detection.

## 5. Discussion

### 5.1 Practical significance of the research results

The ECL-RTDETR model proposed in this paper systematically addresses the common problems in rice disease detection tasks, such as high computational complexity, insufficient real-time performance, and degraded detection accuracy under small targets and complex backgrounds. Experimental results demonstrate that, compared with the original RT-DETR model, ECL-RTDETR maintains high detection precision while significantly reducing the number of parameters and computational complexity, as well as greatly improving the inference speed. This indicates that by introducing lightweight and dynamic modeling mechanisms into the backbone network, feature upsampling, and neck network, efficient and accurate recognition of agricultural disease targets can be achieved without relying on ultra-large-scale models.

From a practical application perspective, rice diseases usually exhibit characteristics such as small scale, irregular morphology, fine texture, and high susceptibility to illumination and occlusion. ECL-RTDETR improves the efficiency of the feature extraction stage via EfficientViT, enhances the representation of lesion edges and fine-grained features through CARAFE upsampling, and strengthens the model's adaptability to target feature variations using the dynamic convolution mechanism of LDConv, making the model more suitable for rice disease detection in complex field environments. Therefore, this study provides a feasible technical solution for efficient and low-cost automatic disease monitoring in smart agriculture scenarios.

### 5.2 Analysis of model limitations

Based on further analysis of the test set visualization results and misdetected samples, although ECL-RTDETR achieves satisfactory overall performance, it still has certain limitations. First, under extreme illumination conditions (e.g., strong direct sunlight or severe shadow occlusion), the contrast between some lesion regions and the background is significantly reduced, which may still lead to missed or false detections in individual samples. Second, since the dataset in this paper mainly covers three common types of rice diseases, the detection performance for rare diseases with limited samples or more complex morphological variations still needs further verification.

In addition, although the introduction of EfficientViT and LDConv reduces the overall computational cost to a certain extent, compared with extremely lightweight models (e.g., some YOLO-nano structures), ECL-RTDETR still has room for further compression in terms of parameters and FLOPs. This suggests that in future research, it is necessary to explore strategies such as model structure pruning, quantization, or knowledge distillation while maintaining detection precision, so as to meet the requirements of more resource-constrained application scenarios.

## 5.3 Performance and challenges in practical agricultural applications

In practical agricultural production, rice disease detection systems often need to be deployed on unmanned aerial vehicles (UAVs), mobile terminals, or agricultural edge computing devices, which impose high requirements on model real-time performance, stability, and power consumption. ECL-RTDETR achieves a good balance between detection speed and model size, giving it the potential for practical deployment in agricultural scenarios, especially suitable for tasks such as UAV field inspection, real-time field monitoring, and rapid disease screening over large farmland areas.

When deploying the ECL-RTDETR model on agricultural edge devices (e.g., UAV-borne computing platforms, field sensing terminals, or embedded edge computing nodes), certain practical challenges may still be encountered. On the one hand, edge devices are usually limited by computing resources, storage capacity, and power budgets. Although the proposed model has been effectively optimized in terms of parameters and computational complexity, further model compression and acceleration may still be required on some low-power or computationally constrained platforms. On the other hand, actual field operations may be affected by factors such as limited network bandwidth, reduced input resolution, and real-time processing of continuous video streams, which impose higher demands on the inference latency and stable operation of the model.

Furthermore, in real-world application environments, the acquisition conditions of rice disease images are highly uncertain, including variations in shooting height, viewing angles, increased background complexity, and complex meteorological conditions, all of which may affect the generalization performance of the model. Therefore, future research can further integrate multi-source information (e.g., multispectral data or temporal features) and conduct more targeted model acceleration and deployment optimization for agricultural edge devices, so as to further improve the stability and practicality of the model in large-scale agricultural production environments.

In summary, although the ECL-RTDETR model proposed in this paper has made certain progress in edge deployment, there is still room for further optimization. Future research can be carried out in the following directions: First, lightweight technologies such as model pruning, quantization, and knowledge distillation can be combined to further reduce the model parameter count and computational complexity, so as to adapt to edge devices with more limited resources; second, hardware-aware neural architecture search (NAS) methods can be explored to achieve collaborative optimization between the model structure and the specific deployment platform; in addition, multimodal data (such as multispectral or temporal information) can be integrated to improve the robustness and generalization ability of the model in complex agricultural environments; finally, inference acceleration optimization (such as TensorRT deployment) can be performed for UAV or embedded platforms to further reduce latency and improve practical application performance.

## Author contributions

**Conceptualization:** Yaojun Zhang.

**Data curation:** Ying Xiong.

**Funding acquisition:** Yaojun Zhang.

**Investigation:** changqiang shen, Ying Xiong.

**Methodology:** Yaojun Zhang, changqiang shen.

**Software:** Yaojun Zhang, changqiang shen.

**Supervision:** Ying Xiong.

**Validation:** Ying Xiong.

**Writing – original draft:** Yaojun Zhang.

**Writing – review & editing:** changqiang shen.

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
