## [Decision Letter · Decision Letter 0]

5 Feb 2026

PONE-D-25-64010Research on an Improved RT-DETR-Based Model for Rice Disease DetectionPLOS One

Dear Dr. shen,

Thank you for submitting your manuscript to PLOS ONE. After careful consideration, we feel that it has merit but does not fully meet PLOS ONE’s publication criteria as it currently stands. Therefore, we invite you to submit a revised version of the manuscript that addresses the points raised during the review process. Please see the comments of the reviewers at the bottom of this email.

We look forward to receiving your revised manuscript.

Kind regards,

Xiaoyong Sun

Academic Editor

PLOS One

Journal Requirements:

3. In the online submission form, you indicated that “We are willing to provide the data upon reasonable request.”

Reviewers' comments:

Reviewer's Responses to Questions

**Comments to the Author**

1. Is the manuscript technically sound, and do the data support the conclusions?

Reviewer #1: Yes

Reviewer #2: Yes

Reviewer #3: Yes

2. Has the statistical analysis been performed appropriately and rigorously? 

Reviewer #1: I Don't Know

Reviewer #2: Yes

Reviewer #3: Yes

3. Have the authors made all data underlying the findings in their manuscript fully available?

Reviewer #1: No

Reviewer #2: Yes

Reviewer #3: Yes

4. Is the manuscript presented in an intelligible fashion and written in standard English?

Reviewer #1: Yes

Reviewer #2: Yes

Reviewer #3: Yes

5. Review Comments to the Author

Reviewer #1: 1. The image quality is low. It is strongly recommended to improve the resolution or provide higher-quality images to ensure better clarity.

2. A discussion section should be added to analyze the implications of the results, limitations, and real-world applications of the model.

3. The reference formatting is inconsistent. Ensure uniformity according to the journal’s guidelines for a more professional presentation.

4. Add numbering to the section titles to improve the manuscript’s structure and make it easier to navigate.

5. Provide the full form of abbreviations like "ViT" and "MSA" when they first appear in the text, followed by their abbreviations.

6. While Table 5 presents ablation experiments for EfficientViT, there are no such experiments for other optimization modules. Please provide an explanation or add corresponding experiments.

7. Provide more comparative visual results to better demonstrate the effectiveness of the proposed method. For example, Figure 4, which visualizes the experimental results, has very few examples, with only three images included.

Reviewer #2: The paper proposes an improved RT-DETR-based model, ECL-RTDETR, for rice disease detection. The overall logic of the paper is coherent, the experimental design is relatively detailed, and the visual analysis supports the conclusions well. However, there is room for improvement regarding the explanation of method sources in comparative experiments, the presentation of certain figures, and the discussion of model limitations. Below are some suggestions for revision:

1. The paper compares YOLOv11n and YOLOv12n models in Table 7 and Figure 11. Given the rapid iteration of the YOLO series and the existence of versions released by different maintenance teams, it is suggested that the authors explicitly state the specific sources (e.g., GitHub repositories) of these versions in the text to ensure the reproducibility of the experiments and the fairness of the comparison.

2. In Sections 2.2 and 2.3, the authors introduce CARAFE and LDConv, respectively. It is recommended to add an explanation as to why these two modules are the optimal choices specifically for the "rice disease" scenario. The current description focuses more on the general advantages of the modules and lacks a close integration with the specific agricultural application context of this study (e.g., how they address specific features of rice lesions).

3. In the network architecture diagram in Figure 5, the orientation and layout of some text appear slightly crowded. Adjustments are suggested to improve readability.

4. Figure 11 includes 8 models and 6 metrics, resulting in overly dense lines, making it difficult to intuitively distinguish the differences in some metrics (such as Size and Params). It is suggested to optimize the chart design or present the radar chart in more effective coordination with the data in Table 7.

5. Although the paper demonstrates the advantages of the improved model over the baseline through Figures 12 and 13, as an application-oriented study, it is recommended to add an analysis of failure cases where the ECL-RTDETR model still falls short. For instance, under what extreme lighting or occlusion conditions does the new model fail? This would help to evaluate the robustness of the model more objectively.

6. When introducing LDConv in [Sec sec005], it is suggested to supplement the specific parameter settings used in the experiments so that readers can understand its actual impact on the computational load.

7. It is suggested to briefly discuss in the Discussion section the potential challenges the ECL-RTDETR model might face when deployed on actual agricultural edge devices, which is also an important consideration for the practical implementation of smart agriculture.

Reviewer #3: This study introduces an enhanced RT-DETR-based mode for rice disease detection. Below are my concerns.

1. Many figures are blurring.

2. The visual results with the competitors (YOLO8, YOLO11, YOLO12 and RTDETR-L) should be provided.

3. Your FLOPs and Network parameters are not the best.

6. PLOS authors have the option to publish the peer review history of their article (what does this mean?). If published, this will include your full peer review and any attached files.

Reviewer #1: No

Reviewer #2: No

Reviewer #3: No

---

## [Author Response · Author response to Decision Letter 1]

28 Feb 2026

We sincerely thank the anonymous reviewers for their valuable comments and constructive suggestions on our manuscript. These comments are highly appreciated and have been of great help in revising and improving the paper. They also provide important guidance for our research. After carefully considering all the reviewers’ remarks, we have made substantial revisions to the manuscript and hope that it can now be accepted for publication.

Reviewer 1:

1. The image quality is low. It is strongly recommended to improve the resolution or provide higher-quality images to ensure better clarity.

Author Response:

We would like to thank the reviewers for their valuable suggestions. We have reprocessed and replaced all illustrations in the manuscript, using uniformly higher-resolution original images and vectorized plotting. In particular, we have optimized the clarity and layout of key experimental result figures (e.g., Figure 5, Figure 11, Figures 12–15). The revised figures maintain good readability even when magnified, significantly improving the overall visual presentation of the manuscript.

2. A discussion section should be added to analyze the implications of the results, limitations, and real-world applications of the model.

Author Response:

We thank the reviewers for their constructive suggestions. Following this comment, we have added a new "Discussion" section after [Sec sec020] of the paper, in which we systematically analyze the following contents:

(1) the practical significance of the ECL-RTDETR model in rice disease detection;

(2) the limitations of the current model under scenarios such as extreme illumination and small-sample disease types;

(3) the potential advantages and applicable conditions of the model in practical agricultural applications.

The relevant content has been clearly marked in the revised manuscript.

3. The reference formatting is inconsistent. Ensure uniformity according to the journal’s guidelines for a more professional presentation.

Author Response:

We thank the reviewers for their careful review and valuable suggestions. We have thoroughly checked all references in the manuscript and uniformly adjusted them in strict accordance with the journal’s formatting requirements, including author format, title capitalization, journal name style, conference paper format, volume, issue, page numbers, and duplicate references. The revised reference list has been updated in the manuscript to improve the overall standardization and presentation of the paper.

4. Add numbering to the section titles to improve the manuscript’sstructure and make it easier to navigate.

Author Response:

We thank the reviewers for their suggestions. We have added uniform numbering to all sections and subsections throughout the manuscript (e.g., 1, 2.1, 3.5.2, etc.), which further optimizes the paper structure and facilitates readers in quickly locating and accessing relevant content.

5. Provide the full form of abbreviations like "ViT" and "MSA" when they first appear in the text, followed by their abbreviations.

Author Response:

We thank the reviewers for their careful review. We have supplemented the full names of abbreviations such as "ViT (Vision Transformer)" and "MSA (Multi-Head Self-Attention)" at their first occurrence in the manuscript, and used the abbreviations consistently thereafter to improve the clarity and standardization of the presentation.

6.While Table 5 presents ablation experiments for EfficientViT, there are no such experiments for other optimization modules. Please provide an explanation or add corresponding experiments.

Author Response:

We thank the reviewers for raising this important concern. We understand that the comment mainly focuses on the completeness and clarity of explaining the contribution of each optimization module to the model performance.

In fact, this paper does not only conduct ablation analysis on the EfficientViT module. The experimental design of Table 5 aims to compare the overall influence of different backbone networks on the performance of RT-DETR. Therefore, this table is mainly used to evaluate the advantages of EfficientViT as the backbone network in terms of accuracy, speed, and computational complexity, rather than the step-by-step ablation of all optimization modules.

Regarding the CARAFE upsampling operator and LDConv dynamic convolution module, a systematic ablation analysis is presented in [Sec sec017] "Ablation Experiments on Optimization Modules", with the corresponding experimental results fully provided in Table 6. By gradually introducing EfficientViT, CARAFE, and LDConv modules, this section quantitatively analyzes the specific effects of each module on detection accuracy, inference speed, the number of parameters, and FLOPs.

To avoid ambiguity for readers, we have further strengthened the logical distinction between Table 5 and Table 6 in the revised manuscript. We have clearly indicated that Table 5 corresponds to the "Backbone Network Comparison Experiments" and Table 6 corresponds to the "Optimization Module Ablation Experiments", and added supplementary descriptions in the relevant sections to improve the completeness and readability of the experimental design.

7. Provide more comparative visual results to better demonstrate the effectiveness of the proposed method. For example, Figure 4, which visualizes the experimental results, has very few examples, with only three images included.

Author Response:

We thank the reviewers for their suggestion. In response to this comment, we have supplemented the revised manuscript with more representative detection and Grad-CAM visualization results, covering different disease types, scale variations, and complex background scenarios, to more comprehensively verify the effectiveness and robustness of the proposed method.

Reviewer #2:

1. The paper compares YOLOv11n and YOLOv12n models in Table 7 and Figure 11. Given the rapid iteration of the YOLO series and the existence of versions released by different maintenance teams, it is suggested that the authors explicitly state the specific sources (e.g., GitHub repositories) of these versions in the text to ensure the reproducibility of the experiments and the fairness of the comparison.

Author Response:

We thank the reviewers for their valuable suggestions. In response to the comment regarding the clarity of the sources of the YOLOv11n and YOLOv12n versions, we have supplemented the specific implementation sources and version information of the models used in the revised manuscript. The YOLOv11n and YOLOv12n models employed in this paper are both implemented based on the official Ultralytics GitHub repository (https://github.com/ultralytics/ultralytics.git). In the experimental setup section, the paper further specifies the detailed implementation information adopted: all models retain the default network structure configurations, and the comparative experiments are conducted under a unified experimental environment with consistent training parameters to ensure the reproducibility of the experimental results and the fairness of the comparisons. The relevant content has been added and explained in [Sec sec013] “Experimental Environment”.

2. In Sections 2.2 and 2.3, the authors introduce CARAFE and LDConv, respectively. It is recommended to add an explanation as to why these two modules are the optimal choices specifically for the "rice disease" scenario. The current description focuses more on the general advantages of the modules and lacks a close integration with the specific agricultural application context of this study (e.g., how they address specificfeatures of rice lesions).

Author Response:

We thank the reviewers for their valuable suggestions. Following this comment, we have added explanations tailored to the rice disease detection scenario in “Sections 2.2 and 2.3” (with further supplements in [Sec sec006]) of the revised manuscript, aiming to strengthen the connection between module selection and the specific agricultural detection task.

In the revised content, considering the characteristics of rice disease targets—such as small lesion sizes, blurred edges, irregular shapes, as well as frequent illumination variations and leaf occlusion in field environments—we further elaborate the advantages of the “CARAFE upsampling operator” in preserving fine-grained edge and texture information of lesions, and the role of the “LDConv dynamic convolution module” in improving the model’s adaptability to varying lesion morphologies and complex backgrounds via its adaptive sampling mechanism.

Through the above supplements, this paper more clearly explains “why the CARAFE and LDConv modules are suitable for rice disease detection” at the method design level, thereby enhancing the rationality and application specificity of the model structure selection. The corresponding revisions have been clearly marked in the revised manuscript.

3. In the network architecture diagram in Figure 5, the orientation and layout of some text appear slightly crowded. Adjustments are suggested to improve readability.

Author Response:

We thank the reviewers for their valuable suggestions. In response to the comment, we have “rearranged and optimized the network architecture diagram in Figure 5”, adjusting the layout and spacing of some text to avoid overlap and overcrowding, thereby improving the overall readability and visual clarity. The relevant revisions have been updated in the revised manuscript.

4. Figure 11 includes 8 models and 6 metrics, resulting in overly dense lines, making it difficult to intuitively distinguish the differences in some metrics (such as Size and Params). It is suggested to optimize the chart design or present the radar chart in more effective coordination with the data in Table 7.

Author Response:

We thank the reviewers for their valuable suggestions. Regarding the issue of dense lines in the radar chart due to the large number of models in Figure 11, we have optimized the visualization method without changing the experimental data. By visually weakening non-critical models and highlighting the display effect of key models, the differences in indicators such as model size and number of parameters have become clearer, while the complete comparative information has been preserved.

5. Although the paper demonstrates the advantages of the improved model over the baseline through Figures 12 and 13, as an application-oriented study, it is recommended to add an analysis of failure cases where the ECL-RTDETR model still falls short. For instance, under what extreme lighting or occlusion conditions does the new model fail? This would help to evaluate the robustness of the model more objectively.

Author Response:

We thank the reviewers for their valuable suggestions. In accordance with this comment, we have added an analysis of failure cases of ECL-RTDETR under extreme illumination and occlusion conditions in the “Discussion section ([Sec sec021])” of the revised manuscript. In addition, we have clearly stated in the captions of “Figure 12 and Figure 13” that the results shown are typical successful detection cases, so as to evaluate the robustness of the model more objectively.

6. When introducing LDConv in [Sec sec005], itis suggested to supplement the specific parameter settings used in the experiments so that readers can understand its actual impact on the computational load.

Author Response:

We thank the reviewers for their valuable suggestions. In response to this comment, we have supplemented the “specific parameter settings of the LDConv module used in the experiments” in “[Sec sec006]” of the revised manuscript, including the convolution kernel size, stride, and the structure of the dynamic offset branch, to help readers better understand the actual impact of this module on the computational load.

Meanwhile, the supplementary explanation emphasizes that LDConv adopts a lightweight design and only introduces a small number of extra parameters. Its influence on the overall model parameters and FLOPs has been verified in the subsequent ablation experiments and overall performance comparison results. The relevant revisions have been clearly marked in the revised manuscript.

7. It is suggested to briefly discuss in the Discussion section the potential challenges the ECL-RTDETR model might face when deployed on actual agricultural edge devices, which is also an important consideration for the practical implementation of smart agriculture.

Author Response:

We thank the reviewers for their valuable suggestions. In response to this comment, we have supplemented **[Sec sec024] (Discussion)** of the revised manuscript with the potential challenges faced when deploying the ECL-RTDETR model on practical agricultural edge devices, including limited computing resources, power consumption constraints, and the impact of complex data acquisition conditions on real-time inference performance. Meanwhile, combined with the smart agriculture application scenario, we have briefly discussed the further optimization directions of the model on edge devices (such as model compression and deployment acceleration), so as to enhance the coverage of practical application issues and the depth of discussion in the paper. The relevant revisions have been clearly marked in the revised manuscript.

Reviewer #3: This study introduces an enhanced RT-DETR-based mode for rice disease detection. Below are my concerns.

1. Many figures are blurring.

Author Response:

We thank the reviewers for their valuable suggestions. We have reprocessed and replaced all illustrations in the manuscript, using uniformly high-resolution original images and vectorized plotting. We also optimized the clarity and layout of key experimental result figures, including Figure 5, Figure 11, and Figures 12–15. The revised figures maintain good readability even when magnified, which significantly improves the overall visual quality of the paper.

2. The visual results with the competitors (YOLO8, YOLO11, YOLO12 and RTDETR-L) should be provided.

Author Response:

We thank the reviewers for their valuable suggestions. In accordance with your comments, we have added a comparative analysis of the visualization detection results on the test set between the proposed model and YOLOv8, YOLOv11, YOLOv12, and RT-DETR models. The relevant comparison results are presented in “Figure 12” of the revised manuscript and described in detail in “[Sec sec019]”. The visualization comparison shows that the proposed “ECL-RTDETR” model outperforms the baseline models in terms of target integrity, localization accuracy, and adaptability to complex backgrounds, which further validates the effectiveness of the proposed method. Thank you again for your constructive comments.

3. Your FLOPs and Network parameters are not the best.

Author Response:

We thank the reviewers for their comments. We understand the concern regarding the comparison between the proposed method and lightweight YOLO-series models.

Indeed, some lightweight YOLO models (such as YOLOv8n, YOLOv11n, and YOLOv12n) present certain advantages in terms of FLOPs and parameter count, but their design objectives are mainly targeted at extremely low-computation scenarios in general object detection.

To address the common challenges in rice disease detection, including small targets, blurred lesion edges, and complex background interference, the proposed ECL-RTDETR maintains controllable FLOPs and parameter scale while significantly improving the modeling capability for fine-grained features and dynamic morphological variations by introducing modules such as EfficientViT, CARAFE, and LDConv.

Experimental results demonstrate that, compared with various lightweight YOLO models, ECL-RTDETR achieves clear advantages in detection accuracy and stability under complex scenarios, while still retaining inference speed that meets the requirements of practical agricultural applications.

Therefore, although ECL-RTDETR does not have the minimal FLOPs or parameter count, its computational scale is consistent and reasonable with the performance improvement achieved. It better satisfies the practical demands of the rice disease detection task, which requires both high precision and real-time performance.

---

## [Decision Letter · Decision Letter 1]

1 Apr 2026

PONE-D-25-64010R1Research on an Improved RT-DETR-Based Model for Rice Disease DetectionPLOS One

Dear Dr. shen,

Thank you for submitting your manuscript to PLOS ONE. After careful consideration, we feel that it has merit but does not fully meet PLOS ONE’s publication criteria as it currently stands. Therefore, we invite you to submit a revised version of the manuscript that addresses the points raised during the review process.  Please submit your revised manuscript by May 16 2026 11:59PM. If you will need more time than this to complete your revisions, please reply to this message or contact the journal office at plosone@plos.org. Please include the following items when submitting your revised manuscript:

We look forward to receiving your revised manuscript.

Kind regards,

Xiaoyong Sun

Academic Editor

PLOS One

Journal Requirements:

**Additional Editor Comments:**

Please see the comments of reviewers at the bottom of this email.

Reviewers' comments:

Reviewer's Responses to Questions

**Comments to the Author**

1. If the authors have adequately addressed your comments raised in a previous round of review and you feel that this manuscript is now acceptable for publication, you may indicate that here to bypass the “Comments to the Author” section, enter your conflict of interest statement in the “Confidential to Editor” section, and submit your "Accept" recommendation.

Reviewer #1: All comments have been addressed

Reviewer #2: All comments have been addressed

Reviewer #3: All comments have been addressed

2. Is the manuscript technically sound, and do the data support the conclusions?

Reviewer #1: Yes

Reviewer #2: Yes

Reviewer #3: Yes

3. Has the statistical analysis been performed appropriately and rigorously? 

Reviewer #1: Yes

Reviewer #2: Yes

Reviewer #3: Yes

4. Have the authors made all data underlying the findings in their manuscript fully available?

Reviewer #1: Yes

Reviewer #2: Yes

Reviewer #3: No

5. Is the manuscript presented in an intelligible fashion and written in standard English?

Reviewer #1: Yes

Reviewer #2: Yes

Reviewer #3: Yes

6. Review Comments to the Author

Reviewer #1: 1. In Table 5, the FPS values for the RTDETR-R101 and RTDETR-Co models are 3.6 and 9.5, respectively, which are relatively low compared to the baseline RTDETR-L's 10.6 FPS. It is suggested to briefly explain the reasons for this in the main text to help readers better understand these results.

2. The discussion on the challenges of edge deployment in [Sec sec024] is well addressed. If feasible optimization directions or technical approaches for the future could be briefly mentioned at the end of this section, it would provide readers with more forward-looking guidance and further strengthen the practical value of the paper.

Reviewer #2: The authors have made reasonable revisions in response to the review comments. The paper can be considered for acceptance.

Reviewer #3: This study proposes ECL-RTDETR, an enhanced RT-DETR–based rice disease detection model. The authors have addressed my concerns. My decision is Accept.

7. PLOS authors have the option to publish the peer review history of their article (what does this mean?). If published, this will include your full peer review and any attached files.

Reviewer #1: No

Reviewer #2: No

Reviewer #3: No

---

## [Author Response · Author response to Decision Letter 2]

8 Apr 2026

Thank you for your valuable comments and suggestions on our manuscript. We have carefully considered each of your remarks and have made the necessary revisions to improve the quality and clarity of our work. Below, please find our detailed responses to each of your comments, along with the specific changes we have implemented in the manuscript. We appreciate your time and effort in reviewing our work.

6. Review Comments to the Author

Reviewer #1:

1. In Table 5, the FPS values for the RTDETR-R101 and RTDETR-Co models are 3.6 and 9.5, respectively, which are relatively low compared to the baseline RTDETR-L's 10.6 FPS. It is suggested to briefly explain the reasons for this in the main text to help readers better understand these results.

Response: Thank you for your valuable comments. We have added a brief explanation in [Sec sec016] of the manuscript to clarify the reasons for the relatively low frame rates of the RTDETR-R101 and RTDETR-Co models. Specifically, the deeper architecture of ResNet-101 and the more complex feature extraction mechanisms in ConvNeXtV2 significantly increase the computational cost (e.g., higher floating-point operations and parameter counts), resulting in slower inference speeds compared to the baseline model RTDETR-L. This explanation has been incorporated into the revised manuscript to improve the clarity of the presentation.

2. The discussion on the challenges of edge deployment in [Sec sec024] is well addressed. If feasible optimization directions or technical approaches for the future could be briefly mentioned at the end of this section, it would provide readers with more forward-looking guidance and further strengthen the practical value of the paper.

Response: Thank you for your valuable suggestions. We have added a brief discussion at the end of [Sec sec024], outlining potential future research directions. These directions include model compression technologies (e.g., pruning, quantization, and knowledge distillation), hardware-aware optimization, neural architecture search, and deployment acceleration strategies for edge devices. This supplement enhances the practicality and forward-looking nature of this paper.

Reviewer #2: The authors have made reasonable revisions in response to the review comments. The paper can be considered for acceptance.

Response: Thank you!

Reviewer #3: This study proposes ECL-RTDETR, an enhanced RT-DETR–based rice disease detection model. The authors have addressed my concerns. My decision is Accept.

Response: Thank you!

---

## [Editor Report · Decision Letter 2]

27 Apr 2026

Research on an Improved RT-DETR-Based Model for Rice Disease Detection

PONE-D-25-64010R2

Dear Dr. shen,

We’re pleased to inform you that your manuscript has been judged scientifically suitable for publication and will be formally accepted for publication once it meets all outstanding technical requirements.

Kind regards,

Xiaoyong Sun

Academic Editor

PLOS One

sunx1@sdau.edu.cn
---

## [Editor Report · Acceptance letter]

PONE-D-25-64010R2

PLOS One

Dear Dr. shen,

I'm pleased to inform you that your manuscript has been deemed suitable for publication in PLOS One. Congratulations! Your manuscript is now being handed over to our production team.

Kind regards,

on behalf of

Dr. Xiaoyong Sun

Academic Editor

PLOS One